# Content of soil organic carbon and labile fractions depend on local combinations of mineral phase characteristics

Malte Ortner[1], Michael Seidel[2], Sebastian Semella[2,3], Thomas Udelhoven[4], Michael Vohland[2], Sören Thiele-Bruhn[1]

[1]Soil Science Department, University of Trier, Trier, 54296, Germany
[2]Geoinformatics and Remote Sensing, Institute for Geography, Leipzig University, Leipzig, 04103, Germany
[3]present address: Deutsches Biomasseforschungszentrum GmbH, 04347 Leipzig, Germany
[4]Department of Remote Sensing & Geoinformatics, University of Trier, Trier, 54296, Germany

**Correspondence:** Malte Ortner (ortner@uni-trier.de), Sören Thiele-Bruhn (thiele@uni-trier.de)

**Abstract.** Soil organic matter (SOM) is an indispensable component of terrestrial ecosystems. Soil organic carbon (SOC) dynamics are influenced by a number of well-known abiotic factors such as clay content, soil pH or pedogenic oxides. These parameters interact with each other and vary in their influence on SOC depending on local conditions. To investigate the latter, the dependence of SOC accumulation on parameters and parameter combinations was statistically assessed that vary on a local scale depending on parent material, soil texture class and land use. To this end, topsoils were sampled from arable and grassland sites in southwestern Germany at four regions with different soil parent material. Principal component analysis (PCA) revealed a distinct clustering of data according to parent material and soil texture that varied largely between the local sampling regions, while land use explained PCA results only to a small extent. The PCA clusters were differentiated into total clusters that contain the entire dataset or major proportions of it and local clusters representing only a smaller part of the dataset. All clusters were analyzed for the relationships between SOC concentrations (SOC %) and mineral phase parameters in order to assess specific parameter combinations explaining SOC and its labile fractions hot water-extractable C (HWEC) and microbial biomass C (MBC). Analyses were focused on soil parameters that are known as possible predictors for the occurrence and stabilization of SOC (e.g. fine silt plus clay and pedogenic oxides). Regarding the total clusters, we found significant relationships, by bivariate models, between SOC, its labile fractions HWEC and MBC and the applied predictors. Yet partly low explained variances indicated the limited suitability of bivariate models. Hence, mixed effect models were used to identify specific parameter combinations that significantly explain SOC and its labile fractions of the different clusters. Comparing measured and mixed effect models-predicted SOC values revealed acceptable to very good regression coefficients ($R^2 = 0.41$-$0.91$) and low to acceptable root mean square error (RMSE = 0.20-0.42 %). Thereby, the predictors and predictor combinations clearly differed between models obtained for the whole data set and the different cluster groups. At a local scale, site specific combinations of parameters explained the variability of organic carbon notably better, while the application of total models to local clusters resulted in less explained variance and a higher RMSE. Independent from that, the explained variance by marginal fixed effects decreased in the order SOC > HWEC > MBC, showing that labile fractions depend less on soil properties but presumably more on processes such as organic carbon input and turnover in soil.

## 1 Introduction

Soil as an inherent part of terrestrial ecosystems acts as a major regulator of the organic carbon (OC) cycle especially through the function of OC storage (Heimann and Reichstein, 2008; Scharlemann et al., 2014). Hence, it is of utmost relevance and a focus of ongoing research to define models and parameter sets that best describe and predict soil organic carbon (SOC) contents of soils. Further it is required to identify the drivers for SOC storage at different scales and sites to adapt the management of soils. Overall, the relevance of parameters for quantification of SOC is often described by bivariate relationships (Hassink et al., 1993; Barré et al., 2017). Yet, SOC and its potential sequestration by formation of organo-mineral associations depends on combinations and interactions of several environmental factors or soil properties, so that the number of multivariate applications to estimate the accumulation of SOC is increasing (Hobley et al., 2015; Heinze et al., 2018).

In addition to total SOC, its labile subfractions such as hot water extractable carbon (HWEC) or microbial biomass carbon (MBC) are more and more recognized as fast reacting SOC pools in order to analyse carbon dynamics in soils (Weigel et al., 2011; Lal, 2016). Both fractions were selected because they are methodically clearly defined compared to other fractions, e.g. particulate organic carbon that is not uniformly defined, either by size or by density (Christensen, 2001; Lützow et al., 2007). Furthermore, it was hypothesized and aimed to show that HWEC and MBC are not closely correlated with each other, and thus deliver different information. The HWEC is known as a measure of the bioavailable and mineralizable fraction of SOC (Spohn and Giani, 2011; Heller and Zeitz, 2012). The MBC is a quantitative measure of the microbial community that plays an indispensable role for the turnover of SOC. Additionally, labile carbon fractions such as MBC quantitatively dominate in short-term turnover processes, while changes in SOC will only become significant over periods of decades. Therefore, MBC is expedient to explain SOC dynamics (Liang et al., 2017). Determination of HWEC and MBC, allows to get a representative measure of the labile SOC pool. Labile carbon fractions were recently simulated (Wieder et al., 2015; Zhang et al., 2021) but compared to SOC they were less considered in the past (Liddle et al., 2020).

It is well known that factors such as climate, topography, vegetation, parent material and time are major factors influencing contents and storage of SOC (Jenny, 1941). Accordingly, large scale (often national or continental) surveys often include geographical properties, vegetation types, general forms of land use as well as climatic site conditions to explain the variability of SOC (Wiesmeier et al., 2014; Gray et al., 2015). Consequently, vegetation and anthropogenic influence by land use and land use changes are essential factors to model SOC accumulation and dynamics (Poeplau and Don, 2013; Dignac et al., 2017). The relevance of the parent material for SOC sequestration and stocks was discussed for sites and small landscapes of a few km² (Barré et al., 2017; Angst et al., 2018) as well as for large areas on the scale of regions or countries (Wiesmeier et al., 2013; Vos et al., 2019). The potential influence of parent material on SOC is mostly considered by parameters of soil mineralogy and texture (Herold et al., 2014). Factors such as climate, topography, parent material, vegetation or land use are well suited to explain the variability of SOC at larger scales or at landscapes with a high variability concerning these factors. In contrast, for smaller, local study areas or rather uniform areas with a low factor variability an inclusion of these factors as variables is less suitable (Wiesmeier et al., 2019).

In addition to these general factors, further parameters describing the soil composition in a more specific way, become relevant at regional or local scale setting boundaries for SOC accumulation, e.g. by the formation of organo-mineral associations. For an identification of SOC variations due to site specific characteristics selected parameters are used which are mostly known as indicators for stabilization of SOC such as content of fine silt, clay and pedogenic oxides or microbial parameters such as microbial biomass and amino sugars (Angst et al., 2018; Quesada et al., 2020). There are indications that for the explanation of SOC variability on a local to regional scale soil parameters (e.g., pedogenic oxides, texture fractions) instead of factors (e.g., parent material or climate) are especially suitable. Models based on soil parameters also allow to identify possible drivers of SOC stabilization while using the above mentioned general factors would not deliver a satisfying result (Wiesmeier et al., 2019; Adhikari et al., 2020).

Organo-mineral associations are highly relevant for stabilization and accumulation of SOC and its labile fractions (Lützow et al., 2006). It is well known that the different mineral particle size classes vary in their ability to interact with SOC, forming organo-mineral associations (Arrouays et al., 2006; Lützow et al., 2007). On one hand coarse particle size fractions such as sand, coarse silt

(cSilt) and medium silt (mSilt) contribute less to interactions between SOC and the mineral phase while on the other hand fine silt (fSilt) and clay dominate such interactions (Ludwig et al., 2003). In addition, the mineral composition of the fine fraction, i.e. types of clay minerals and pedogenic oxides, is relevant for the interactions of SOC with the mineral phase (Kleber et al., 2015; Porras et al., 2017). Especially iron and aluminum oxides interact with SOC leading to its sequestration (Mikutta et al., 2006). Stabilization of SOC is further enhanced by multivalent cations such as $Ca^{2+}$ and $Mg^{2+}$ going along with higher soil pH (Kaiser et al., 2012; O'Brien et al., 2015). Covering on one hand all quantitative relevant cations and on the other hand being an overall measure of soils sorptive properties, the effective cation exchange capacity (ECEC) provides an overall measure to model cation impact on SOC storage (Kaiser et al., 2012; O'Brien et al., 2015). Rock fragments (soil skeleton) contribute only little to SOC storage (Poeplau et al., 2017). Anyhow, the fraction of rock fragments is considered as a relevant parameter to assess SOC accumulation due to a potential saturation effect in soils with a high rock fragment content in consequence of a disproportionately high input of organic matter in the fine soil fraction (Bornemann et al., 2011).

Consequently, understanding SOC as a dynamic equilibrium of heterogeneous compounds with distinct relationships to various components of the soil mineral phase (Lehmann and Kleber, 2015) implies that SOC accumulation is best described and predicted by a variety of soil mineral phase parameters instead of a single predictor. Thereby combinations of parameters or factors can differ according to the considered scale. Consequently, multivariate approaches better explain the SOC variability (Heinze et al., 2018; Liddle et al., 2020) compared to bivariate linear regression models that are often unsuited at the level of local and regional soilscapes (Jian-Bing et al., 2006). The latter especially applies for studies that are limited to a single specific location or only contain a limited number of categorical variables or estimated soil parameters (Liddle et al., 2020). On the other hand, predictions based on total models, based on the largest part of the dataset, are often less site-specific and thus can possibly lead to a weaker quantification of SOC at certain sites.

Consequently, it is required to determine parameter sets to estimate SOC and its labile fractions HWEC and MBC at a regional or landscape scale. It is necessary to identify predictor parameters and categorical environmental factors that are able to predict SOC as well as its labile fractions by using models based on local and total datasets. Differences regarding the relevance of a predictor in local vs. total models have to be identified to boost model performance and to fit adequate datasets using the best set of parameters for the prediction of SOC at the investigated location. This overall aim was investigated in this study using a dataset from four local agricultural areas in the greater region of Trier (each with a size of 5-10 km²), thus with similarity in the global factors but distinct local properties such as parent material, soil texture and land use. Regarding the composition of the soil mineral phase the four local areas differ among each other, but as a total dataset they represent a broad range of soil properties typical for soils in temperate regions. Therefore, the dataset enables to verify whether the total dataset is able to cover the local variability of SOC and its labile fractions. Objectives of this study were, (i) to determine best fitting factors and parameter combinations, based on identified differences in soil properties, that explain the variability in SOC and its labile fractions HWEC and MBC. (ii) It was aimed to determine the suitability of local models in comparison to total models to achieve an improved quantification of SOC, HWEC and MBC for local landscapes with distinct properties. To this end, bivariate linear regression, principal component analysis (PCA) and mixed effect models were used in order to find out whether total models or local models are better fitting. (iii) It was assessed if local datasets show a distinct combination of significantly contributing predictor parameters compared to other local datasets and the entire dataset.

## 2 Material and Methods

### 2.1 Study area

The study was conducted in the greater area of Trier in southwestern Germany (Fig. 1). Bulk samples from topsoil horizons, i.e. 0-25 cm for arable and 0-15 cm for grassland soils, were taken in spring 2017 and 2018 from 199 agricultural sites used as arable land (150) and grassland (49). Similar numbers of samples were taken from four regional areas with different parent materials. Parent

materials were Devonian clay schist (DCS, n= 50), Luxemburg sandstone (LBS, n= 50), sandy dolomitic limestone (DLS, n= 50)
from the Muschelkalk, and Permian siltstone and fine sandstone (PSS, n= 49) from the Rotliegend (Wagner et al., 2011). Across
the different parent materials, a similar proportion of samples were taken at sites under arable or grassland management. Climatic
conditions in the greater area of Trier are classified as warm-temperate, fully humid with warm summer temperate (Cfb) (Kottek et
al., 2006). According to the German Weather service (DWD) mean annual precipitation is 784 mm and mean annual temperature is
9.8°C. Investigated sites were dominated by the soil groups Regosol and Cambisol. The main cultivated crop plants are wheat,
barley, triticale, maize and rapeseed.

**Fig. 1.** Study area in the greater Trier region; sampling sites at the four regions with different parent material are indicated, i.e.
Devonian clay schist (DCS), sandy dolomitic limestone (DLS) from the Muschelkalk, Luxemburg sandstone (LBS), and Permian
siltstone and fine sandstone (PSS) from the Rotliegend (©GeoBasis-DE).

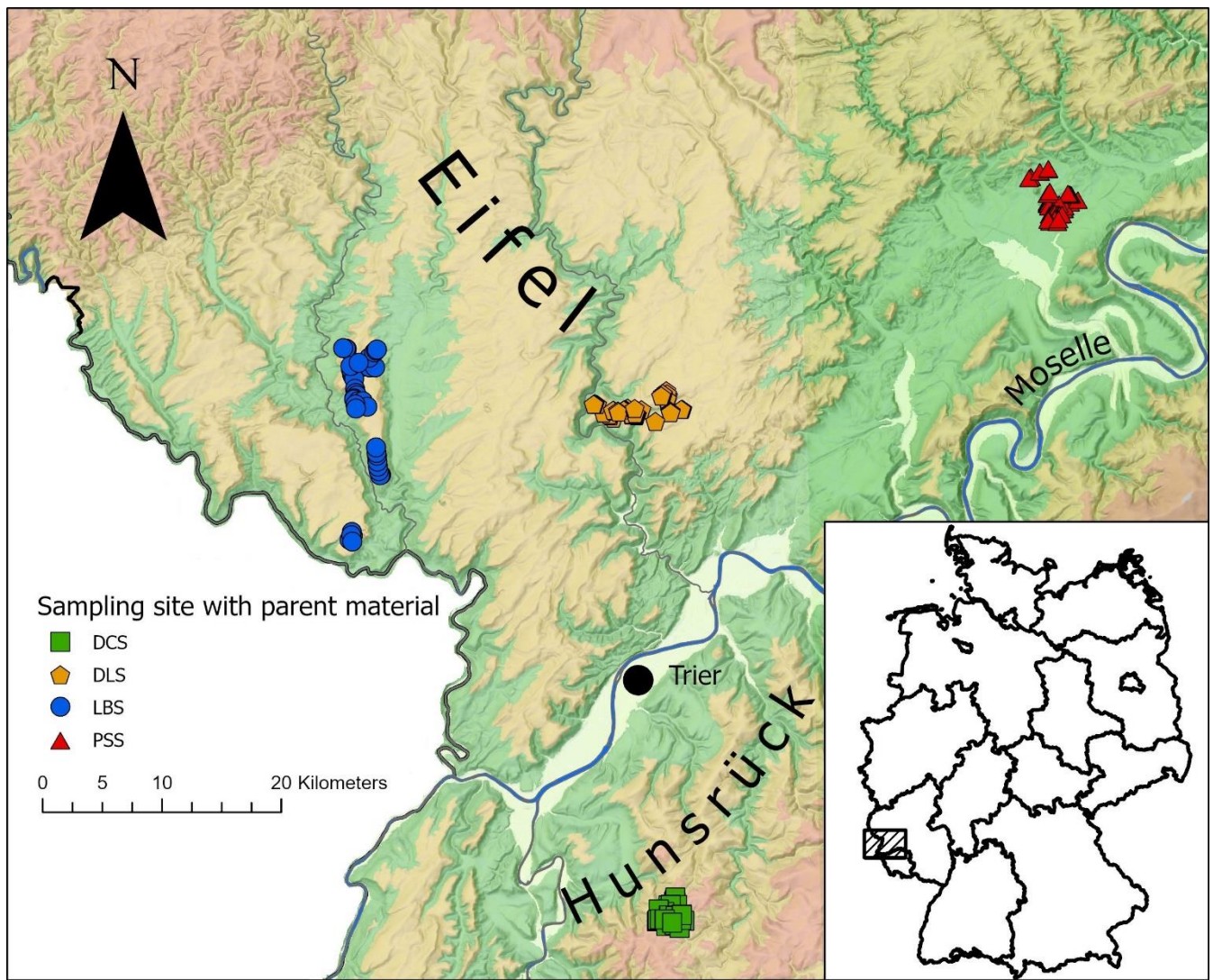

**2.2 Analysis of soil properties**

Samples were sieved < 2 mm and the stone content (> 2 mm) was determined gravimetrically. Each sample was split and stored at
134 -20°C on one hand and air-dried on the other hand for subsequent biological and chemical soil analysis, respectively. Soil pH was
135 measured in 0.01 M $CaCl_2$ solution using a pH/Con 340i glass electrode (WTW GmbH, Weilheim). Particle size distribution was
136 determined by a combination of wet sieving and pipette method according to Blume et al. (2011). Dithionite-citrate extractable Fe
($Fe_d$) was measured according to Mehra and Jackson (1958). To this end, 2 g air-dry soil were extracted with a mixture of 1 g sodium

dithionite, 40 ml sodium citrate and 10 ml $NaHCO_3$. Oxalate extractable Fe and Al ($Fe_o$, $Al_o$) were determined according to

Schwertmann (1964). For extraction, 1 g air-dry soil was shaken for 2 h in the dark in 50 ml $NH_4^+$-oxalate (pH 3) and filtered

afterwards. Extraction for the determination of the effective cation exchange capacity (ECEC) was conducted using 1 M $NH_4Cl$.

Elemental analyses for pedogenic oxides and ECEC (Na, K, Fe, Mn, Al, Ca, Mg) were done using atomic absorption spectrometry

(Varian AA240 FS Fast Sequential Atomic Absorption Spectrometer; Darmstadt, Germany).

For estimation of total carbon (TC) and nitrogen soil was dried at 105°C, grinded and measured by an Elemental Analyser EA3000

Series (HEKAtech GmbH, Wegberg). For carbonate containing soils the inorganic carbon (IC) was determined following carbonate

destruction using phosphoric acid at a temperature of 100°C (IC Kit combined with Elemental Analyser EA3000 Series, HEKAtech

GmbH, Wegberg). SOC content was calculated as the difference of TC and IC. HWEC and hot water extractable nitrogen (HWEN)

were determined following Körschens et al. (1990), using a Gerhardt Turbotherm TT 125 (Gerhardt, Bonn, Germany) for extraction

of 10 g soil with distilled water (50 ml) at 100°C for 1 h. After extracts cooled down 1 ml of 0.2 M $MgSO_4$ was added and samples

were centrifuged at 1476 g for 10 minutes. Microbial biomass was estimated by using chloroform fumigation extraction according

to Joergensen (1995) with 0.01 M $CaCl_2$. Extracts of HWEC, HWEN, microbial biomass carbon (MBC) and nitrogen (MBN) were

analysed with a TOC-VCPN analyser (Shimadzu, Duisburg, Germany). For MBC and MBN correction factors kEC = 0.45 and kEN

= 0.4 respectively, were used (Joergensen, 1996; Joergensen and Mueller, 1996). Soils were stored frozen prior to analysis for MBC

and MBN because freezing does not affect the microflora (Stenberg et al., 1998). Soil respiration was measured according to

Heinemeyer et al. (1989). Following a week of incubation at room temperature (20 °C), 25 g dry matter equivalent of sieved field

moist soil were weighted in a tube that was flushed with 200 mL $min^{-1}$ of $CO_2$-free, humid air for 24 hours. Evolved $CO_2$ was

determined in one-hour intervals after the soil passage using an infrared gas analyser (ADC 225 MK3, The Analytical Development,

Hoddesdon, England).

## 2.3 Data analysis

Principal component analysis (PCA) was carried out to identify clusters within the dataset. For that purpose, 24 parameters

describing the mineral phase as well as SOM were included (Table 1). To conduct the PCA applied variables were log transformed,

centered and scaled to achieve standardized and comparable variables. Ellipses were defined by 95 % of the confidence interval

according to Fox and Weisberg (2019), The cluster of clayey soils was not included in the analysis due to a small number of samples

(n = 5). Using single predictors, linear regressions were performed to identify significant impact of mineral phase parameters (e.g.

$Fe_o$ [g $kg^{-1}$] or fSilt plus clay [%]) on SOC, HWEC and MBC for the entire dataset as well as for the identified clusters. Residues of

the bivariate linear regressions were checked for normality. Mixed effect models were determined for the entire dataset and for

identified clusters. To this end, selected soil properties of the mineral phase ($Fe_{d-o}$ [g $kg^{-1}$], $Fe_o$ [g $kg^{-1}$], $Al_o$ [g $kg^{-1}$], sand [%], cSilt

plus mSilt [%], fSilt plus clay [%], $(Ca + Mg)_{ECEC}$ [mmolc $kg^{-1}$], stones [%] and pH) were used as fixed effect while, 'parent

material', 'soil texture group' or 'land use' were used as random effect. In general, as random effects only categorical variables

were selected, while for the fixed effects variable mineral phase parameters were selected. Parent material as a random effect

includes the four different soil parent materials that dominate at the four sampling sites. For the soil texture group as random effect

four levels were applied (sandy, silty, clayey and loamy soils). The additional implementation of the soil texture groups was done

to consider the potential different intercepts of the specific groups. Land use as random effect comprised the two management

practices arable and grassland. Restricted maximum likelihood was applied as estimation procedure for the mixed effect models. At

the beginning, all selected soil properties were included in each model. Stepwise removal of the least significant parameters was

conducted until all properties included in the models significantly contributed to SOC, HWEC or MBC. Additionally, the relevance

of variables was visualized by the mean values of the clusters multiplied with their coefficient received from the mixed effect

models. All parameters involved as fixed parameter in the mixed effect models were checked for collinearity. To avoid collinear

behaviour of the soil texture related parameters either 'sand' or 'coarse silt plus medium silt' (cSilt plus mSilt) were used for model

development. The two models received were compared by their Akaike information criterion (AIC) using ANOVA to identify the

best model. Furthermore, ECEC was excluded from mixed effect models to avoid overfitting due to collinearity with $(Ca+Mg)_{ECEC}$.

Residuals of models were examined for homoscedasticity and normality. In case these criteria were not fulfilled, the response

variable was square root transformed to achieve variance homogeneity and normality. For the mixed effect models a marginal $R^2$

($R^2_{marg}$) and conditional $R^2$ ($R^2_{con}$) coefficient was estimated according to Nakagawa and Schielzeth (2013). Thereby $R^2_{marg}$ examines

the explained variance of the fixed effects while $R^2_{cond}$ tests the variance including the effect of the random effects. The root mean

square error (RMSE) was estimated as a measure of the model performance. Both $R^2$ and RMSE were used for a comparative

assessment of different models rather than for an absolute valuation. For the mixed effect models, RMSE was estimated based on

the comparison of predicted and measured values. To transfer the mixed effect models of a total dataset to a local dataset, predictions

were conducted applying total models to local datasets. The predicted values of SOC; HWEC and MBC received from the different

mixed effect models were compared with the measured values using bivariate linear regressions. This yielded $R^2$ and RMSE as

measures of goodness. All data are shown as mean ($\pm$ SE) if not indicated otherwise. Statistical significance is indicated with $*p <$

0.05, $**p < 0.01$ and $***p < 0.001$. Statistical analyses were carried out using the R statistical package version 4.1.1 (R Core Team,

2021).

**Table 1.** Soil properties in agricultural topsoils for the complete dataset and defined group levels according to parent material, land use and soil texture class. Values are means ± SD. * indicates mineral phase related parameters which were applied in bivariate models.

| | Dataset (n=199) | DCS (n= 50) | LBS (n=50) | DLS (n=50) | PSS (n=49) | Sandy soils (n=54) | Loamy soils (n=98) | Silty soils (n=42) | Arable (n=150) | Grassland (n=49) |
|---|---|---|---|---|---|---|---|---|---|---|
| SOC [%] | 1.94 ± 0.87 | 3.03 ± 0.78 | 1.61 ± 0.39 | 1.92 ± 0.49 | 1.17 ± 0.33 | 1.41 ± 0.45 | 2.08 ± 0.87 | 2.08 ± 0.76 | 1.82 ± 0.73 | 2.29 ± 1.11 |
| Nitrogen [%] | 0.20 ± 0 10 | 0.33 ± 0.08 | 0.14 ± 0.03 | 0.19 ± 0.04 | 0.13 ± 0.04 | 0.13 ± 0.03 | 0.22 ± 0.10 | 0.21 ± 0.08 | 0.19 ± 0.08 | 0.23 ± 0.12 |
| Hydrogen [%] | 0.56 ± 0.29 | 0.94 ± 0.25 | 0.33 ± 0.07 | 0.59 ± 0.15 | 0.38 ± 0.12 | 0.33 ± 0.08 | 0.64 ± 0.31 | 0.63 ± 0.22 | 0.57 ± 0.29 | 0.54 ± 0.27 |
| Oxygen [%] | 3.77 ± 1.93 | 6.15 ± 0.97 | 2.45 ± 0.63 | 3.87 ± 2.00 | 2.56 ± 0.70 | 2.18 ± 0.53 | 4.23 ± 1.63 | 4.67 ± 2.46 | 3.62 ± 1.82 | 4.24 ± 2.14 |
| HWEC [µg g⁻¹] | 753 ± 322 | 1071 ± 353 | 661 ± 163 | 732 ± 214 | 545 ± 252 | 570 ± 199 | 813 ± 319 | 782 ± 276 | 669 ± 231 | 1010 ± 410 |
| HWEN [µg g⁻¹] | 99.4 ± 42.0 | 130 ±35.5 | 78.9 ± 28.2 | 107 ± 37.2 | 80.4 ± 43.8 | 70.4 ± 31.0 | 106 ± 39.9 | 116 ± 40.3 | 93.8 ± 39.9 | 116 ± 43.5 |
| MBC [µg g⁻¹] | 247 ± 143 | 325 ± 159 | 130 ± 42.1 | 320 ± 118 | 209 ± 117 | 123 ± 47.2 | 271 ± 132 | 322 ± 119 | 205 ± 93.0 | 377 ± 186 |
| MBN [µg g⁻¹] | 41.2 ± 23.5 | 53.5 ± 25.3 | 22.6 ± 8.78 | 50.8 ± 23.2 | 37.1 ± 17.7 | 22.9 ± 10.2 | 44.5 ± 22.1 | 52.3 ± 21.7 | 35.5 ± 18.3 | 58.5 ± 28.5 |
| Respiration [µg CO₂-C (g dry matter h)] | 0.26 ± 0.11 | 0.29 ± 0.11 | 0.21 ± 0.05 | 0.30 ± 0.12 | 0.22 ± 0.10 | 0.20 ± 0.07 | 0.27 ± 0.11 | 0.28 ± 0.10 | 0.23 ± 0.09 | 0.32 ± 0.13 |
| MBC/SOC | 1.36 ± 0.71 | 1.05 ± 0.31 | 0.89 ± 0.61 | 1.75 ± 0.72 | 1.73 ± 0.62 | 0.97 ± 0.51 | 1.42 ± 0.67 | 1.71 ± 0.82 | 1.23 ± 0.66 | 1.74 ± 0.70 |
| SOC/N | 11.7 ± 2.13 | 10.5 ± 0.98 | 13.7 ± 2.20 | 11.9 ± 2.11 | 10.7 ± 1.25 | 12.9 ± 2.58 | 11.1 ± 1.55 | 11.5 ± 2.06 | 11.6 ± 2.24 | 12.0 ± 1.76 |
| HWE-C/N | 9.90 ± 4.98 | 9.76 ± 2.33 | 11.1 ± 5.02 | 8.60 ± 2.85 | 10.2 ± 7.61 | 11.64 ± 6.77 | 9.66 ± 4.45 | 8.20 ± 2.15 | 9.80 ± 5.64 | 13.3 ± 6.40 |
| MB-C/N | 7.41 ± 2.57 | 7.62 ± 2.66 | 7.54 ± 3.36 | 7.83 ± 2.00 | 6.66 ± 1.81 | 7.02 ± 3.17 | 7.57 ± 2.44 | 7.55 ± 2.01 | 7.36 ± 2.79 | 7.55 ± 1.68 |
| IC [%] | 0.37 ± 1.18 | - | - | 1.43 ± 1.98 | - | - | 0.12 ± 0.62 | 1.36 ± 2.04 | 0.39 ± 1.24 | 0.29 ± 0.98 |
| *pH | 4.98 ± 0.89 | 4.78 ± 0.61 | 4.70 ± 0.72 | 5.89 ± 0.77 | 5.47 ± 0.57 | 4.79 ± 0.73 | 5.02 ± 0.76 | 5.46 ± 0.90 | 5.02 ± 0.87 | 4.88 ± 0.87 |
| *ECEC [mmolc kg⁻¹] | 65.6 ± 29.2 | 66.8 ± 21.0 | 38.8 ± 14.4 | 96.7 ± 26.3 | 58.6 ± 15.2 | 40.1 ± 12.9 | 66.6 ± 21.6 | 94.7 ± 28.5 | 65.6 ± 28.6 | 65.5 ± 31.1 |
| *Ca + Mg_ECEC [mmolc kg⁻¹] | 55.7 ± 28.5 | 54.2 ± 21.2 | 30.8 ± 14.3 | 86.4 ± 26.6 | 50.0 ± 13.6 | 32.4 ± 12.9 | 55.5 ± 21.1 | 84.9 ± 28.5 | 54.9 ± 27.6 | 58.3 ± 31.0 |
| *Fe₀ [g kg⁻¹] | 2.34 ± 1.18 | 3.95 ± 0.72 | 1.40 ± 0.40 | 2.24 ± 0.80 | 1.77 ± 0.66 | 1.32 ± 0.32 | 2.69 ± 1.12 | 2.66 ± 1.05 | 2.30 ± 1.14 | 2.49 ± 1.27 |
| *Fe_d-Fe₀ [g kg⁻¹] | 4.57 ± 2.18 | 6.92 ± 2.00 | 3.27 ± 1.18 | 4.50 ± 1.48 | 3.54 ± 1.74 | 2.91 ± 1.15 | 5.22 ± 2.20 | 5.10 ± 2.04 | 4.67 ± 2.23 | 4.27 ± 1.97 |
| *Al₀ [g kg⁻¹] | 1.26 ± 1.13 | 2.98 ± 0.89 | 0.84 ± 0.44 | 0.62 ± 0.30 | 0.61 ± 0.21 | 0.77 ± 0.45 | 1.53 ± 1.25 | 1.10 ± 0.98 | 1.21 ± 1.07 | 1.42 ± 1.28 |
| *Sand [%] | 44.2 ± 23.1 | 26.8 ± 5.80 | 69.1 ± 17.9 | 24.6 ± 8.71 | 57.6 ± 13.5 | 75.1 ± 10.8 | 38.4 ± 13.5 | 21.0 ± 5.09 | 44.8 ± 23.8 | 42.3 ± 20.9 |
| *cSilt+ mSilt [%] | 29.1 ± 13.3 | 30.8 ± 4.53 | 17.6 ± 13.6 | 43.7 ± 6.67 | 23.4 ± 7.97 | 13.5 ± 8.39 | 30.5 ± 7.11 | 45.1 ± 6.59 | 28.8 ± 13.5 | 29.9 ± 12.8 |
| *fSilt + clay [%] | 26.8 ± 12.7 | 42.4 ± 4.95 | 13.2 ± 4.75 | 31.7 ± 5.74 | 19.0 ± 7.01 | 11.5 ± 3.22 | 31.1 ± 10.6 | 33.9 ± 6.67 | 26.4 ± 13.0 | 27.8 ± 11.7 |
| *Stones [%] | 14.3 ± 12.3 | 29.3 ± 8.91 | 6.70 ± 6.51 | 13.1 ± 8.99 | 7.59 ± 8.29 | 6.88 ± 4.52 | 18.0 ± 14.0 | 14.5 ± 10.1 | 15.0 ± 12.7 | 11.9 ± 10.7 |

**3 Results**

**3.1 Soil properties and cluster identification**

The dataset covers soils with broad ranges of 24 parameters and parameter ratios of SOM, soil mineral phase and microbial biomass (Table 1). For example, soil pH ranged from very strongly acidic (pH 3.8) to slightly alkaline (pH 7.4); soil texture varied from sandy to clayey texture. Parent materials essentially influenced characteristics of the mineral phase related parameters such as texture. For example, soils developed from sandy parent material such as LBS had a sandy texture with sand content of up to 91.9 %. Soils developed from DCS and DLS parent material had elevated contents of fine silt plus clay (33.4-53.3 % and 16.7-44.8 %, respectively). Additionally, high contents of pedogenic oxides were found in soils from DCS while ECEC and especially the contents of the polyvalent cations $(Ca+Mg)_{ECEC}$ were high in soils developed from DLS (Table 1). Higher contents of SOC, HWEC and MBC were found for all parent material substrates in grassland soils compared to arable soils (Table 1 and Table S1). For the entire dataset, SOC ranged from 0.38 to 5.32 %, while ranges from 237 to 1889 µg/g and 52.4 to 810 µg/g were determined for HWEC and MBC, respectively. SOC was strongly correlated with HWEC ($R^2 = 0.75$) while the regression with MBC was substantially lower ($R^2 = 0.40$). The dissimilar regressions of SOC with the two labile fractions indicate differences between HWEC and MBC, which was further confirmed by the mediocre regression between HWEC and MBC ($R^2 = 0.55$).

To identify possible local clusters due to different sampling sites, parent material or land use systems within the dataset, PCA was conducted including all 24 soil parameters and parameter ratios (Fig. 2). Principal component (PC) 1 to 3 explained 65 % of the variance and had eigenvalues > 1 (Table 2). Parameters related to the soil mineral phase loaded on all three PCs. Additionally, highest loadings on PC 1 were found for parameters describing the composition of SOM such as content of SOC, nitrogen, hydrogen or oxygen as well as HWEC or MBC. For PC 2 high loadings were further found for parameters related to soil acidity (pH, IC, ECEC, $(Ca+Mg)_{ECEC}$), as well as for SOC and the microbial ratio MBC/SOC. HWEC and respiration further loaded on PC 3 (Table 2). A plot of the first two PCs shows clear clusters that were strongly related to the parent materials according to the different sampling sites (Fig. 2 A). In addition, samples clustered differently when assigned to different soil texture classes (Fig. 2 B). Land use, however, was insufficient to explain separation into different local clusters (Fig. S1). Instead, the land use clusters covered soils from all sampling regions and property combinations, and thus represented total clusters. Compared to the entire dataset or the land use clusters, the identified clusters based on parent material and soil texture covered distinct property ranges of SOC and the mineral phase (Table 1). In contrast to the local clusters, the total cluster according to land use classes showed mostly properties quite similar to the entire dataset. Overall, identified clusters strongly depended on the composition of SOM as well as on specific properties of the soil mineral phase, e.g. texture or soil pH related properties. With a smaller relevance, parameters regarding the characteristics of soil microorganisms separated the dataset into clusters (Table 2).

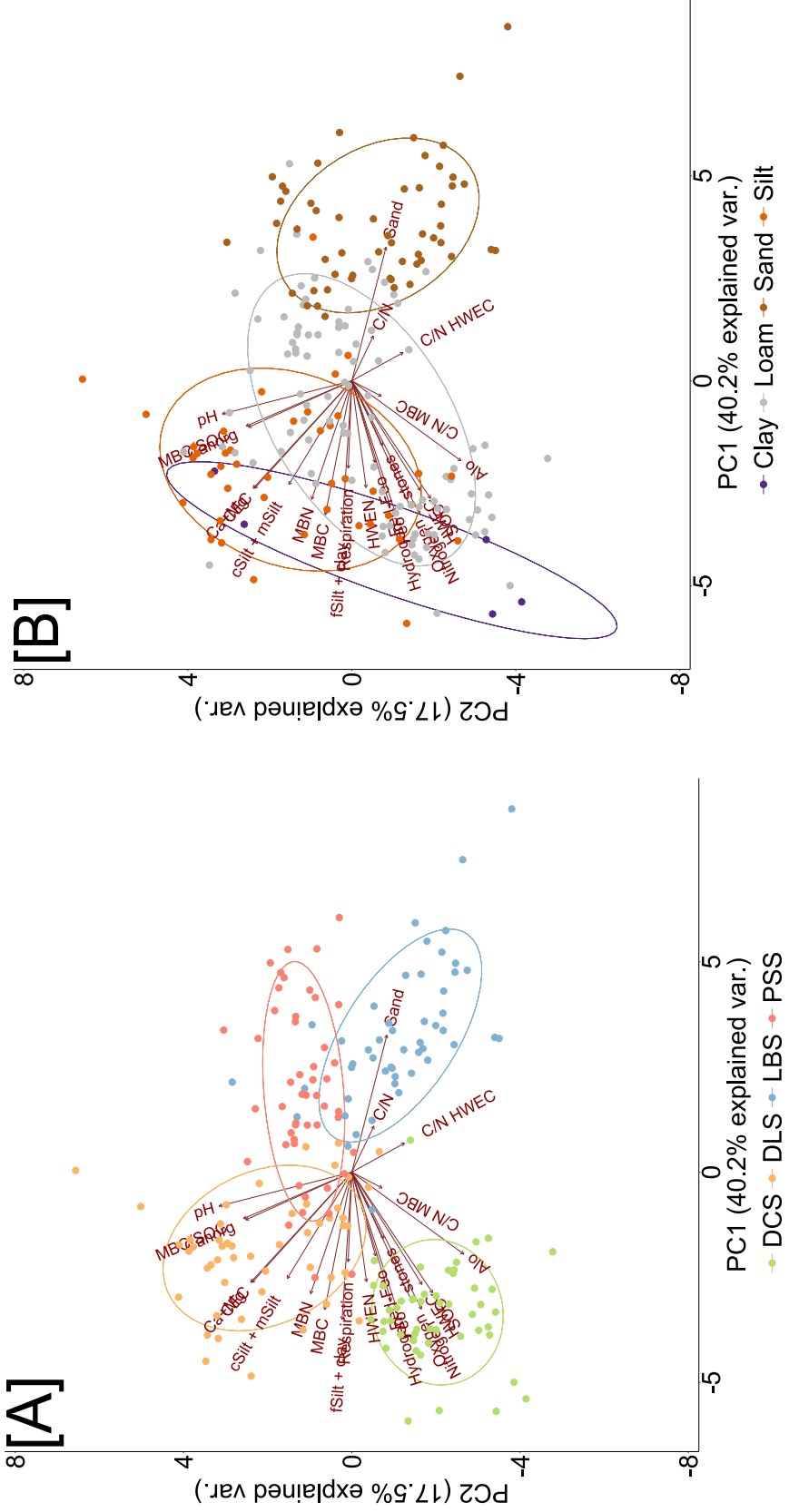

**Fig. 2.** Principal components 1 and 2 with loadings of the variables indicating the clustering of the dataset according to parent material [A] and soil texture [B]. Parent materials are Devonian clay schist (DCS), sandy dolomitic limestone (DLS), Luxemburg sandstone (LBS), and Permian siltstone and fine sandstone (PSS).

b

**Table 2.** Loadings of the variables on the first three principal components.

|  | PC1 | PC2 | PC3 |
|---|---|---|---|
| SOC | -0.24 | -0.24 | -0.19 |
| Nitrogen | -0.27 | -0.21 | -0.04 |
| Hydrogen | -0.26 | -0.12 | 0.17 |
| Oxygen | -0.26 | -0.18 | 0.07 |
| HWEC | -0.22 | -0.21 | -0.36 |
| HWEN | -0.22 | -0.04 | -0.19 |
| MBC | -0.27 | 0.08 | -0.26 |
| MBN | -0.24 | 0.12 | -0.26 |
| Respiration | -0.18 | 0.01 | -0.36 |
| MBC/SOC | -0.09 | 0.33 | -0.12 |
| C/N SOM | 0.09 | -0.07 | -0.36 |
| C/N HWEC | 0.06 | -0.16 | -0.13 |
| C/N MB | -0.03 | -0.09 | 0.04 |
| IC | -0.09 | 0.32 | -0.09 |
| pH | -0.07 | 0.4 | 0.03 |
| ECEC | -0.22 | 0.3 | 0.07 |
| $(Ca+Mg)_{ECEC}$ | -0.22 | 0.3 | 0.06 |
| Feo | -0.27 | -0.13 | 0.12 |
| Fed-Feo | -0.17 | -0.07 | 0.37 |
| Alo | -0.16 | -0.34 | 0.14 |
| Sand | 0.27 | -0.11 | -0.11 |
| cSilt + mSilt | -0.21 | 0.19 | 0.12 |
| fSilt + clay | -0.29 | 0.03 | 0.18 |
| Stones | -0.13 | -0.09 | 0.29 |
| Proportion of Variance | 40.2 | 17.5 | 7.47 |
| Cumulative Proportion | 40.2 | 57.8 | 65.23 |
| Eigenvalue | 9.66 | 4.21 | 1.79 |

**3.2 Bivariate relationships of mineral phase and SOC and its labile fractions**

In order to test whether single parameters are suitable predictors of SOC, HWEC and MBC ten out of 23 parameters describing the properties of the soil mineral phase were selected from the dataset (Table 1, Table 3). Bivariate linear regressions were calculated based on the total dataset (n = 199), for further total clusters (e.g. arable or grassland soils) and the local clusters that were identified in PCA, i.e. subgroups based on the four parent materials and major texture classes (Table 3). Using the complete dataset, highly significant regressions of SOC, HWEC and MBC to most soil mineral phase parameters were found, yet predominantly at a low level of explained variance (Table 3). Compared to the complete dataset substantially different soil parameters explained SOC, HWEC and MBC especially for smaller clusters such as soils from the parent materials DCS or LBS. Yet, clusters comprising large sample numbers, where soil parameters cover broad ranges such as the clusters of loamy, arable or grassland soils, showed significantly contributing parameters that were largely in line with those found as significant for the complete dataset. All clusters differed in their pattern of significant parameters. However, for the complete dataset as well as for the clusters the explained variance decreased from SOC to the labile fractions HWEC and MBC (Fig. 3 and Table 3). Only some properties such as sand. ECEC or $(Ca+Mg)_{ECEC}$ showed for MBC a higher explained variance compared to SOC and HWEC (Table 3). For the entire dataset the content of SOC was best explained by $Al_o$ and $Fe_o$ as predictor parameter ($R^2 =$ 0.58 and 0.56, respectively) while soil texture related properties such as sand or fSilt plus clay explained SOC on a lower level (Table 3). Other determined mineral phase parameters such as cSilt plus mSilt or ECEC explained variance to a negligible extent (Table 3). With lower values for $R^2$, HWEC was explained by similar soil mineral phase parameters, as it was the case for SOC. With $R^2$ of 0.39 and a variance of 0.38 HWEC was best explained by pedogenic oxides ($Fe_o$ and $Al_o$, Table 3). In contrast, the predictors for MBC were quite distinct. Especially parameters related to soil texture such as fSilt plus clay ($R^2 = 0.43$) or sand ($R^2 = 0.45$) better explained the variance of MBC compared to HWEC ($R^2 = 0.27$ and 0.16, respectively). Nevertheless, none of the applied parameters could explain in all cases the complete variance of SOC, HWEC or MBC to a higher extent ($R^2 > 0.75$). Explained variance of SOC and its labile fractions varied strongly between the parent material clusters. In general, the variance in these clusters was explained to a substantially lower extent compared to the whole dataset (Table 3). In most cases, parameters of soil texture and pedogenic oxides correlated significantly with SOC, HWEC and MBC. Additional to these parameters, $(Ca+Mg)_{ECEC}$ was useful to predict SOC and MBC for some parent material clusters (Table 3). Highest values of $R^2$ were reached for the regression between SOC and $Al_o$ and $Fe_o$ (0.47, 0.42) in the cluster DCS and fSilt plus clay (0.37) in the cluster PSS. $R^2$ was even lower in the clusters LBS and DLS with maximum values of 0.21 and 0.20 respectively. Further, the cluster of loamy soils was also best described by parameters representing pedogenic oxides and texture. Much lower $R^2$ were found for the sandy and silty soil clusters with $Al_o$ and texture parameters (sandy) and additionally $Fe_o$ (silty) as best descriptors. While for SOC, HWEC and MBC mostly the same descriptors were found (yet on different level of $R^2$), they were partially different for MBC of the clusters silty and loamy soils.

**Table 3.** Bivariate linear regression coefficient $R^2$ for parameters explaining the variance of SOC [%], HWEC and MBC [µg kg⁻¹] respectively, for soils groups of different parent material, major textural class and land use

| | | Fe$_o$ [g kg⁻¹] | Fe$_d$-Fe$_o$ [g kg⁻¹] | Al$_o$ [g kg⁻¹] | Sand [%] | cSilt + mSilt [%] | fSilt + clay [%] | Stones [%] | ECEC [mmolc kg⁻¹] | (Ca+Mg)$_{ECEC}$ [mmolc kg⁻¹] | pH |
|---|---|---|---|---|---|---|---|---|---|---|---|
| | | | | | | **All samples** | | | | | |
| Dataset n = 199 | SOC | 0.56*** | 0.16*** | 0.58*** | 0.23*** | 0.04** | 0.46*** | 0.24*** | 0.07*** | 0.05** | 0.02 |
| | HWEC | 0.39*** | 0.05** | 0.38*** | 0.16*** | 0.04** | 0.27*** | 0.11*** | 0.03* | 0.02* | 0.06*** |
| | MBC | 0.29*** | 0.07*** | 0.10*** | 0.45*** | 0.29*** | 0.43*** | 0.06*** | 0.29*** | 0.28*** | 0.04** |
| | | | | | | **Land use** | | | | | |
| Arable n = 150 | SOC | 0.51*** | 0.25*** | 0.61*** | 0.23*** | 0.05** | 0.46*** | 0.29*** | 0.09*** | 0.05** | 0.02 |
| | HWEC | 0.37*** | 0.11*** | 0.37*** | 0.18*** | 0.06*** | 0.28*** | 0.17*** | 0.08*** | 0.04* | 0.03* |
| | MBC | 0.25*** | 0.15*** | 0.06*** | 0.64*** | 0.52*** | 0.51*** | 0.12*** | 0.61*** | 0.53*** | 0.21*** |
| Grassland n = 49 | SOC | 0.73*** | 0.08* | 0.73*** | 0.25*** | 0.02 | 0.59*** | 0.44*** | 0.04 | 0.04 | 0.00 |
| | HWEC | 0.67*** | 0.03 | 0.59*** | 0.21*** | 0.02 | 0.47*** | 0.30*** | 0.00 | 0.00 | 0.05 |
| | MBC | 0.54*** | 0.07 | 0.24*** | 0.41*** | 0.13*** | 0.67*** | 0.15** | 0.11* | 0.11* | 0.00 |
| | | | | | | **Parent material** | | | | | |
| DCS n = 50 | SOC | 0.42*** | 0.25*** | 0.47*** | 0.00 | 0.04 | 0.03 | 0.00 | 0.00 | 0.00 | 0.02 |
| | HWEC | 0.17** | 0.24*** | 0.17** | 0.00 | 0.01 | 0.00 | 0.03 | 0.00 | 0.00 | 0.04 |
| | MBC | 0.14** | 0.18** | 0.06 | 0.00 | 0.03 | 0.01 | 0.06 | 0.00 | 0.00 | 0.01 |
| LBS n = 50 | SOC | 0.01 | 0.11* | 0.18* | 0.11* | 0.11* | 0.08* | 0.00 | 0.10* | 0.11* | 0.10* |
| | HWEC | 0.03 | 0.03 | 0.06 | 0.01 | 0.01 | 0.01 | 0.00 | 0.05 | 0.04 | 0.00 |
| | MBC | 0.16** | 0.04 | 0.00 | 0.21*** | 0.19** | 0.21*** | 0.00 | 0.20** | 0.17** | 0.06 |
| DLS n = 50 | SOC | 0.03 | 0.03 | 0.00 | 0.02 | 0.00 | 0.08* | 0.20** | 0.20** | 0.20** | 0.03 |
| | HWEC | 0.07 | 0.05 | 0.00 | 0.00 | 0.00 | 0.00 | 0.06 | 0.04 | 0.04 | 0.02 |
| | MBC | 0.02 | 0.00 | 0.03 | 0.05 | 0.00 | 0.11* | 0.08* | 0.19** | 0.19** | 0.06 |
| PSS n = 49 | SOC | 0.35*** | 0.00 | 0.28*** | 0.36*** | 0.23*** | 0.37*** | 0.04 | 0.30*** | 0.27*** | 0.02 |
| | HWEC | 0.20** | 0.03 | 0.21*** | 0.30*** | 0.29*** | 0.20** | 0.12* | 0.10* | 0.09* | 0.08* |
| | MBC | 0.15** | 0.00 | 0.28*** | 0.44*** | 0.37*** | 0.35*** | 0.02 | 0.16** | 0.17** | 0.10* |
| | | | | | | **Texture** | | | | | |
| Sandy n = 54 | SOC | 0.00 | 0.01 | 0.40*** | 0.18** | 0.19*** | 0.07 | 0.01 | 0.02 | 0.02 | 0.04 |
| | HWEC | 0.00 | 0.06 | 0.29*** | 0.08* | 0.06 | 0.08* | 0.03 | 0.03 | 0.04 | 0.11* |
| | MBC | 0.13** | 0.08* | 0.04 | 0.08* | 0.12* | 0.00 | 0.00 | 0.07 | 0.05 | 0.01 |
| Silty n = 42 | SOC | 0.25*** | 0.02 | 0.33*** | 0.01 | 0.22*** | 0.27*** | 0.00 | 0.03 | 0.03 | 0.01 |
| | HWEC | 0.20** | 0.00 | 0.12* | 0.00 | 0.08 | 0.08 | 0.04 | 0.01 | 0.02 | 0.04 |
| | MBC | 0.01 | 0.06 | 0.00 | 0.16** | 0.00 | 0.07 | 0.05 | 0.17** | 0.19** | 0.06 |
| Loamy n = 89 | SOC | 0.63*** | 0.16*** | 0.70*** | 0.41*** | 0.01 | 0.56*** | 0.36*** | 0.04* | 0.02 | 0.10** |
| | HWEC | 0.36*** | 0.02 | 0.36*** | 0.20*** | 0.01 | 0.24*** | 0.13*** | 0.00 | 0.00 | 0.15*** |
| | MBC | 0.08** | 0.00 | 0.04* | 0.12*** | 0.02 | 0.13*** | 0.00 | 0.14*** | 0.15*** | 0.00 |

**Fig. 3.** Predicted *vs.* measured content of SOC [A], HWEC [B] and MBC [C] for the global (complete) dataset, based on mixed effect models. Parent materials are Devonian clay schist (DCS), sandy dolomitic limestone (DLS), Luxemburg sandstone (LBS), and Permian siltstone and fine sandstone (PSS). RMSE is given in % for SOC and in mg/g for HWEC and MBC.

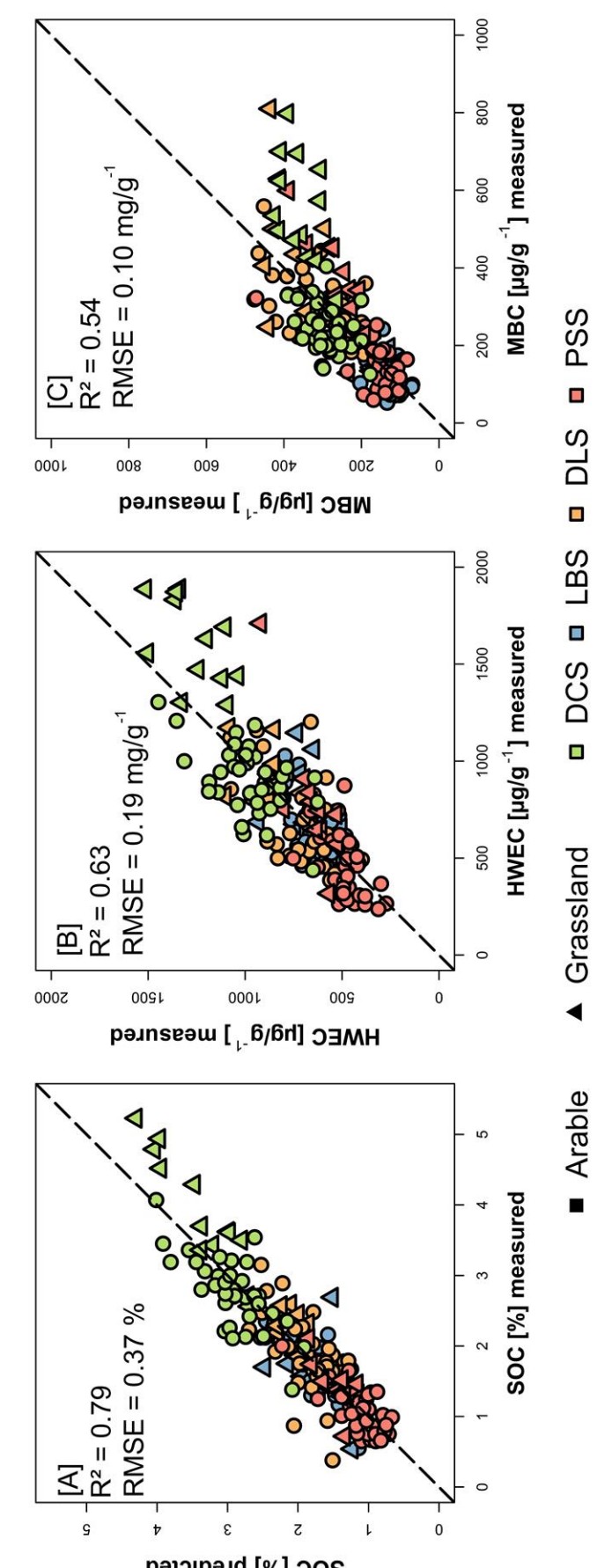

Comprising soils from all identified clusters, the sets of descriptor parameters of the land use clusters were comparable to those

of the total dataset (Table 3). Yet, the variance of SOC and its labile fractions were explained by bivariate linear regressions to

a much higher extent for the total dataset and the clusters of arable soils and especially grassland soils compared to the clusters

based on parent material and texture (Table 3). The clusters of both land use types largely overlapped and contained a similar

proportion of samples from each parent material. Therefore, they can act as total clusters. While SOC was explained by

interactions of numerous different parameters (up to eight) for the distinct factors, less variables showed a significant

contribution to explain the variability of HWEC and MBC (Table 3).

## 3.3 Estimation of SOC and its labile fractions by mixed effect models

Since bivariate linear models mostly explained SOC, HWEC and MBC only to a small extent ($R^2 < 0.5$), mixed effect models

were developed. In these models, mineral phase parameters were applied as fixed effects, and land use, parent material and

texture were used as random effects (Table 4, Fig. 4 and Fig. 5). Variability of SOC, HWEC and MBC were much better

explained than by linear regressions indicating that organic matter depends on complex interactions of several components of

the mineral phase. Based on marginal effects, the mixed effect models explained the variance in most cases in the order SOC >

HWEC > MBC (Fig. 3, Table 4 and Table 5). The mixed effect models reached a higher explained variance and mostly lower

RMSE for SOC ($R^2_{cond} = 0.39\text{-}0.89$, RMSE = $0.21 - 0.42$ %) compared to the bivariate regressions ($R^2 = 0.00\text{-}0.73$, RMSE =

0.27-1.12 %). Data for RMSE are listed in Table SI2. Accordingly, the mixed effect models yielded a higher explained variance

for HWEC and MBC. Representing the explained variance of the fixed effects, the $R^2_{marg}$ revealed for the majority of the clusters

a large explained variance. But even in the cases of low $R^2_{marg}$ several of that clusters had a high $R^2_{con}$. This highlights the

relevance of the random effects (Table 4). Applying different random effects resulted in large differences in $R^2_{cond}$ for some

clusters (e.g., 'sandy soils'). In particular, modelling the labile fractions was more affected by the different random effects,

showing mostly highest $R^2_{cond}$ values if land use was applied as random effect.

Independent from the applied random effect, explained variance increased with sample number and width of the data range of

parameters. Consequently, best model performance was achieved for the complete dataset as well as for the total clusters. Similar

model performance was only found for some local clusters (e.g. DCS), while models for other local clusters such as LBS, DLS

or sandy soils revealed the poorest, yet still better ($R_{cond} \geq 0.39$, RMSE $\leq 0.40$ %) estimates of SOC compared to bivariate

regression models (Table 3, Table 4). In general, applying random effects such as parent material, land use or texture for mixed

effect models led to distinct results for the prediction of SOC, HWEC or MBC (Table 4). For clusters according to land use

variance was explained to a high extent (mean $R^2_{con}$ of 0.66 and 0.77 for cluster of arable soils and grassland, respectively).

Models using parent material or texture as random effect mostly showed minor differences for predictions of SOC, HWEC or

MBC. Anyhow for some local clusters (e.g. DCS, LBS and DLS) distinct results were found. Models using land use as random

effect were partly distinct, though, indicating the different influence of land use on SOC and its labile fractions (Table 4).

The different mixed effects models particularly comprised variables (Fig. 4, Fig. 5) that also proved significant in the bivariate

linear regressions (Table 3). Mineral phase parameters contributed with different significance to the models for SOC, HWEC

and MBC. The SOC and HWEC were primarily explained by pedogenic oxides followed by soil texture related parameters. Not

least, soil acidity specified by pH and $(Ca+Mg)_{ECEC}$ was also relevant. MBC, compared to SOC or HWEC, was better explained

by parameters linked to soil texture. Contribution of the variables, on SOC and its labile fraction was visualized using the mean

values multiplied with their coefficients (Fig. 4, Fig 5). Distinct significant parameter combinations explaining SOC, HWEC

and MBC were also found between the total data set and local clusters (Table 3, Fig. 4 and Fig. 5, SI Table S3). For example,

within the soil texture related clusters pedogenic oxides, $(Ca+Mg)_{ECEC}$, pH and texture parameters were relevant to estimate

SOC, HWEC and MBC (Table 3, Fig. 4 and Fig. 5). Regarding the random effects, applied mixed effect models using parent

material as random effect explained variability of SOC best (Table 4). For MBC and HWEC, however, highest explained

variance were mostly obtained with land use as random effect (Table 4). Only estimates of HWEC for the texture clusters were

better when parent material was used as random effect.

**Table 4.** $R^2_{marg}$ and $R^2_{con}$ of the models for SOC, HWEC, and MBC based on the results of mixed effect models.

| | Land use | | | | | | Parent material | | | | | | Texture | | | | | |
| | SOC | | HWEC | | MBC | | SOC | | HWEC | | MBC | | SOC | | HWEC | | MBC | |
| | $R^2_{marg}$ | $R^2_{con}$ | $R^2_{marg}$ | $R^2_{con}$ | $R^2_{marg}$ | $R^2_{con}$ | $R^2_{marg}$ | $R^2_{con}$ | $R^2_{marg}$ | $R^2_{con}$ | $R^2_{marg}$ | $R^2_{con}$ | $R^2_{marg}$ | $R^2_{con}$ | $R^2_{marg}$ | $R^2_{con}$ | $R^2_{marg}$ | $R^2_{con}$ |
|---|---|---|---|---|---|---|---|---|---|---|---|---|---|---|---|---|---|---|
| Data | 0.74 | 0.76 | 0.48 | 0.65 | 0.39 | 0.78 | 0.65 | 0.78 | 0.47 | 0.59 | 0.57 | 0.59 | 0.76 | 0.76 | 0.56 | 0.59 | 0.51 | 0.57 |
| Arable | | | | | | | 0.62 | 0.77 | 0.37 | 0.55 | 0.72 | 0.72 | 0.71 | 0.71 | 0.5 | 0.51 | 0.69 | 0.71 |
| Grassland | | | | | | | 0.85 | 0.89 | 0.43 | 0.71 | 0.72 | 0.74 | 0.88 | 0.88 | 0.65 | 0.65 | 0.72 | 0.77 |
| DCS | 0.38 | 0.82 | 0.10 | 0.84 | 0.03 | 0.89 | | | | | | | 0.52 | 0.73 | 0.31 | 0.68 | 0.17 | 0.72 |
| LBS | 0.40 | 0.43 | 0.13 | 0.36 | 0.21 | 0.57 | | | | | | | 0.39 | 0.39 | 0.27 | 0.27 | 0.1 | 0.46 |
| DLS | 0.48 | 0.48 | 0.33 | 0.34 | 0.14 | 0.42 | | | | | | | 0.48 | 0.48 | 0.35 | 0.37 | 0.25 | 0.25 |
| PSS | 0.57 | 0.58 | 0.31 | 0.61 | 0.09 | 0.86 | | | | | | | 0.50 | 0.59 | 0.54 | 0.56 | 0.42 | 0.56 |
| Sandy soils | 0.52 | 0.52 | 0.45 | 0.48 | 0.15 | 0.75 | 0.13 | 0.85 | 0.29 | 0.61 | 0.21 | 0.35 | | | | | | |
| Silty soil | 0.69 | 0.69 | 0.65 | 0.65 | 0.39 | 0.47 | 0.43 | 0.73 | 0.33 | 0.86 | 0.45 | 0.45 | | | | | | |
| Loamy soils | 0.75 | 0.81 | 0.47 | 0.67 | 0.19 | 0.75 | 0.55 | 0.76 | 0.49 | 0.56 | 0.41 | 0.41 | | | | | | |
| Mean | 0.57 | 0.65 | 0.36 | 0.57 | 0.20 | 0.69 | 0.54 | 0.80 | 0.40 | 0.65 | 0.52 | 0.54 | 0.61 | 0.65 | 0.45 | 0.52 | 0.41 | 0.58 |

**Fig. 4.** Coefficients of the mixed effect models to predict SOC, multiplied with the mean values of the specific cluster indicating

the impact of the applied variables. Differentiation into clusters and used random factors. Variables are scaled from 0 to 1.

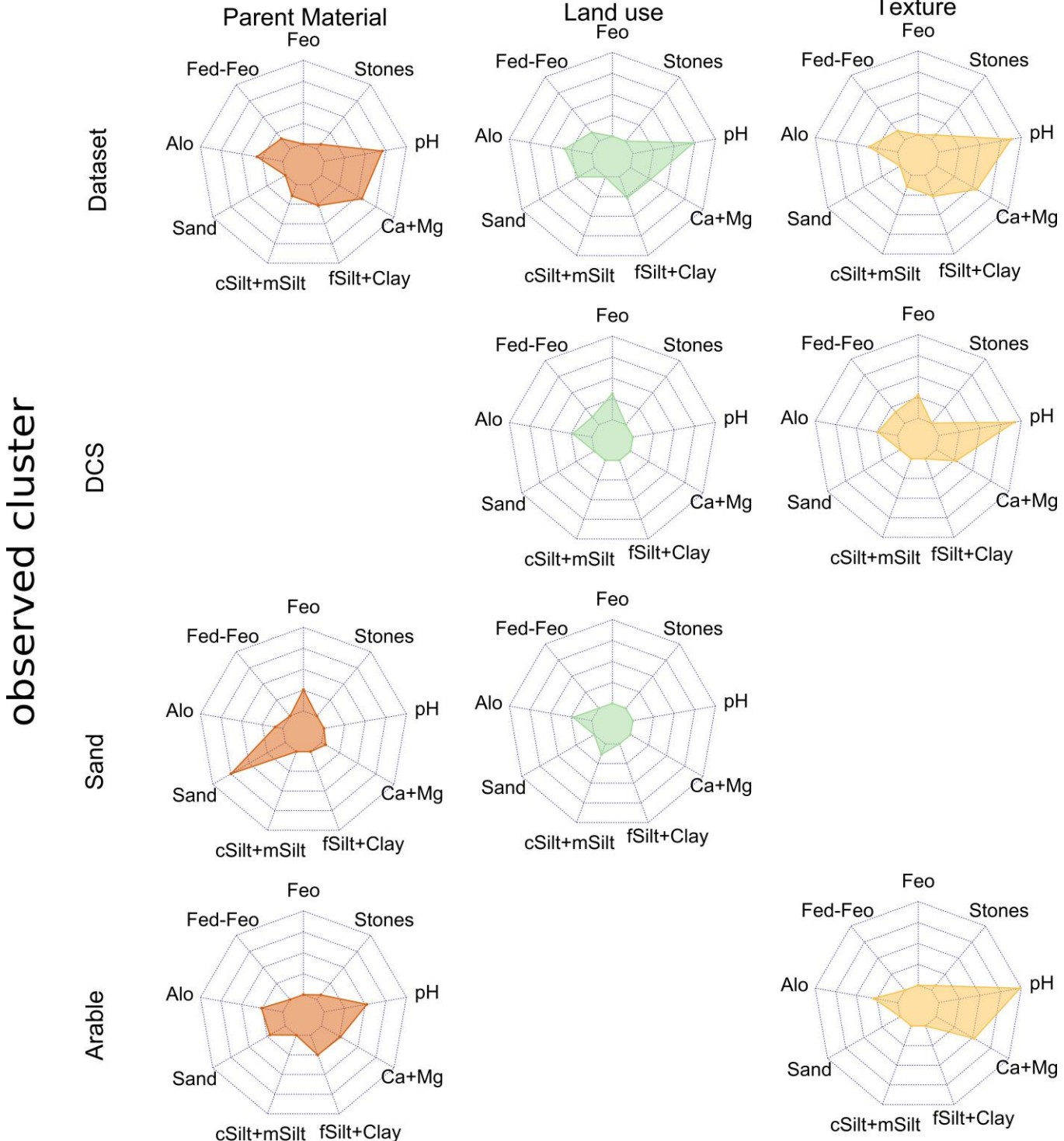

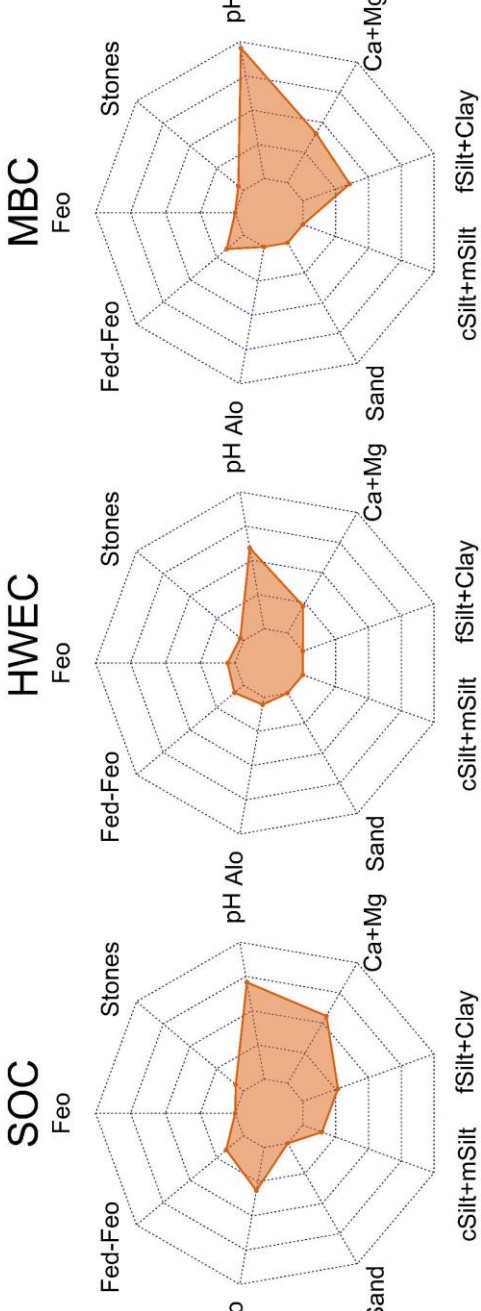

**Fig. 5**. Comparison of the coefficient impact for mixed effect models to predict SOC, HWEC and MBC for the entire dataset by using parent material as random factor. Variables are scaled from 0 to 1.

**3.4 Comparison of total and local explained variability.**

Predictions for SOC, HWEC and MBC were conducted based on the mixed effects models. Subsequent linear regression between measured and predicted data showed a close relationship along the 1:1 prediction line leading to a high explained variance (Fig 3, Table 5). For these regressions the explained variance was mostly similar to $R^2_{con}$. Especially for the total clusters, i.e. the total dataset and data clustered according to arable or grassland land use, best results were obtained. Yet, this was at least partly due to a larger sample size and a broader range of parameter values compared to the various local clusters. Applying a total model for SOC estimation to a smaller local cluster data set clearly revealed an inferior explained variance of the total compared to the local model (Fig. 6). Alongside with decreasing explained variance, RMSE values were mostly increasing if a total model was applied to a local dataset. The higher explained variance of specific local models and parameter combinations was also found for other local clusters (Table 6 and SI Table S4). By transferring a total model to local clusters, the explained variance differed for SOC by up to 20 % while RMSE differed by up to 0.25 %. Also in case a total model was transferred to a local dataset to estimate HWEC or MBC, the explained variance decreased by up to 17 % for HWEC and MBC. The RMSE increased by up to 0.07 and 0.06 mg $g^{-1}$ for HWEC and MBC, respectively.

**Table 5.** $R^2$ and RMSE of the models for prediction of SOC, HWEC, and MBC based on the results of mixed effect models. RMSE is given in % for SOC and in mg/g for HWEC and MBC.

| Sample | | Parent material[+] SOC | HWEC | MBC | Land use[+] SOC | HWEC | MBC | Texture[+] SOC | HWEC | MBC | Mean model prediction $R^2$ |
|---|---|---|---|---|---|---|---|---|---|---|---|
| Dataset | $R^2$ | 0.79* | 0.63* | 0.55* | 0.77* | 0.68* | 0.71* | 0.77* | 0.61* | 0.56* | 0.67* |
| | RMSE | 0.37 | 0.19 | 0.10 | 0.41 | 0.18 | 0.08 | 0.42 | 0.20 | 0.10 | 0.40 % / 0.14 mg $g^{-1}$ |
| *Land use* | | | | | | | | | | | |
| Arable | $R^2$ | 0.80* | 0.59* | 0.70* | | | | 0.72* | 0.53* | 0.71* | 0.68* |
| | RMSE | 0.33 | 0.15 | 0.05 | | | | 0.39 | 0.16 | 0.05 | 0.36 % / 0.10 mg $g^{-1}$ |
| Grassland | $R^2$ | 0.91* | 0.78* | 0.76* | | | | 0.89* | 0.67* | 0.76* | 0.80* |
| | RMSE | 0.33 | 0.19 | 0.09 | | | | 0.36 | 0.24 | 0.09 | 0.35 % / 0.15 mg $g^{-1}$ |
| *Parent Material* | | | | | | | | | | | |
| DCS | $R^2$ | | | | 0.81* | 0.78* | 0.79* | 0.74* | 0.62* | 0.55* | 0.72 |
| | RMSE | | | | 0.34 | 0.17 | 0.07 | 0.40 | 0.22 | 0.11 | 0.37 % / 0.14 mg $g^{-1}$ |
| LBS | $R^2$ | | | | 0.43* | 0.27* | 0.48* | 0.41* | 0.29* | 0.41* | 0.38 |
| | RMSE | | | | 0.30 | 0.14 | 0.03 | 0.30 | 0.14 | 0.03 | 0.30 % / 0.09 mg $g^{-1}$ |
| DLS | $R^2$ | | | | 0.50* | 0.36* | 0.32* | 0.50* | 0.37* | 0.26* | 0.39 |
| | RMSE | | | | 0.35 | 0,17 | 0.10 | 0.35 | 0.17 | 0.10 | 0.35 % / 0.13 mg $g^{-1}$ |
| PSS | $R^2$ | | | | 0.61* | 0.62* | 0.74* | 0.63* | 0.56* | 0.60* | 0.62 |
| | RMSE | | | | 0.21 | 0.16 | 0.06 | 0.20 | 0.16 | 0.07 | 0.21 % / 0.11 mg $g^{-1}$ |
| *Texture* | | | | | | | | | | | |
| Sandy | $R^2$ | 0.79* | 0.61* | 0.28* | 0.54* | 0.51* | 0.58* | | | | 0.55 |
| | RMSE | 0.21 | 0.12 | 0.04 | 0.31 | 0.14 | 0.03 | | | | 0.26 % / 0.08 mg $g^{-1}$ |
| Silty soils | $R^2$ | 0.74* | 0.75* | 0.48* | 0.72* | 0.66* | 0.50* | | | | 0.64 |
| | RMSE | 0.39 | 0.14 | 0.09 | 0.40 | 0.16 | 0.08 | | | | 0.40 % / 0.12 mg $g^{-1}$ |
| Loamy | $R^2$ | 0.83* | 0.59* | 0.41* | 0.81* | 0.66* | 0.64* | | | | 0.66 |
| | RMSE | 0.35 | 0.20 | 0.10 | 0.38 | 0.19 | 0.08 | | | | 0.37 % / 0.14 mg $g^{-1}$ |
| *Mean model prediction* | | | | | | | | | | | |
| Mean | $R^2$ | 0.81 | 0.66 | 0.53 | 0.65 | 0.57 | 0.59 | 0.67 | 0.52 | 0.55 | |
| | RMSE | 0.33 | 0.17 | 0.08 | 0.34 | 0.16 | 0.07 | 0.35 | 0.18 | 0.08 | |

[+]Applied random effect; ˜Not all random effects could applied to this group of clusters because of missing factor levels. *Significant on a level of <0.05

**Table 6.** $R^2$ and RMSE for implementation of the total dataset to local clusters to estimate SOC.

| | | SOC | | | | | |
|---|---|---|---|---|---|---|---|
| | | Parent material | | Land use | | Texture | |
| Sample subgroups | | Cluster specific model | total model to local cluster | Cluster specific model | total model to local cluster | Cluster specific model | total model to local cluster |
| Dataset | $R^2$ | 0.79 | | 0.77 | | 0.77 | |
| | RMSE | 0.37 | | 0.41 | | 0.42 | |
| DCS | $R^2$ | | | 0.81 | 0.69 | 0.74 | 0.65 |
| | RMSE | | | 0.34 | 0.44 | 0.40 | 0.47 |
| LBS | $R^2$ | | | 0.43 | 0.23 | 0.41 | 0.24 |
| | RMSE | | | 0.30 | 0.41 | 0.30 | 0.40 |
| DLS | $R^2$ | | | 0.50 | 0.30 | 0.50 | 0.35 |
| | RMSE | | | 0.35 | 0.42 | 0.35 | 0.41 |
| PSS | $R^2$ | | | 0.61 | 0.57 | 0.63 | 0.57 |
| | RMSE | | | 0.21 | 0.38 | 0.20 | 0.38 |
| Sandy soils | $R^2$ | 0.79 | 0.68 | 0.54 | 0.37 | | |
| | RMSE | 0.21 | 0.26 | 0.31 | 0.36 | | |
| Silty soils | $R^2$ | 0.74 | 0.65 | 0.72 | 0.60 | | |
| | RMSE | 0.39 | 0.45 | 0.40 | 0.48 | | |
| Loamy soils | $R^2$ | 0.83 | 0.83 | 0.81 | 0.79 | | |
| | RMSE | 0.35 | 0.36 | 0.38 | 0.40 | | |
| Arable | $R^2$ | 0.80 | 0.80 | | | 0.72 | 0.73 |
| | RMSE | 0.33 | 0.34 | | | 0.39 | 0.39 |
| Grassland | $R^2$ | 0.91 | 0.87 | | | 0.89 | 0.87 |
| | RMSE | 0.33 | 0.44 | | | 0.36 | 0.47 |

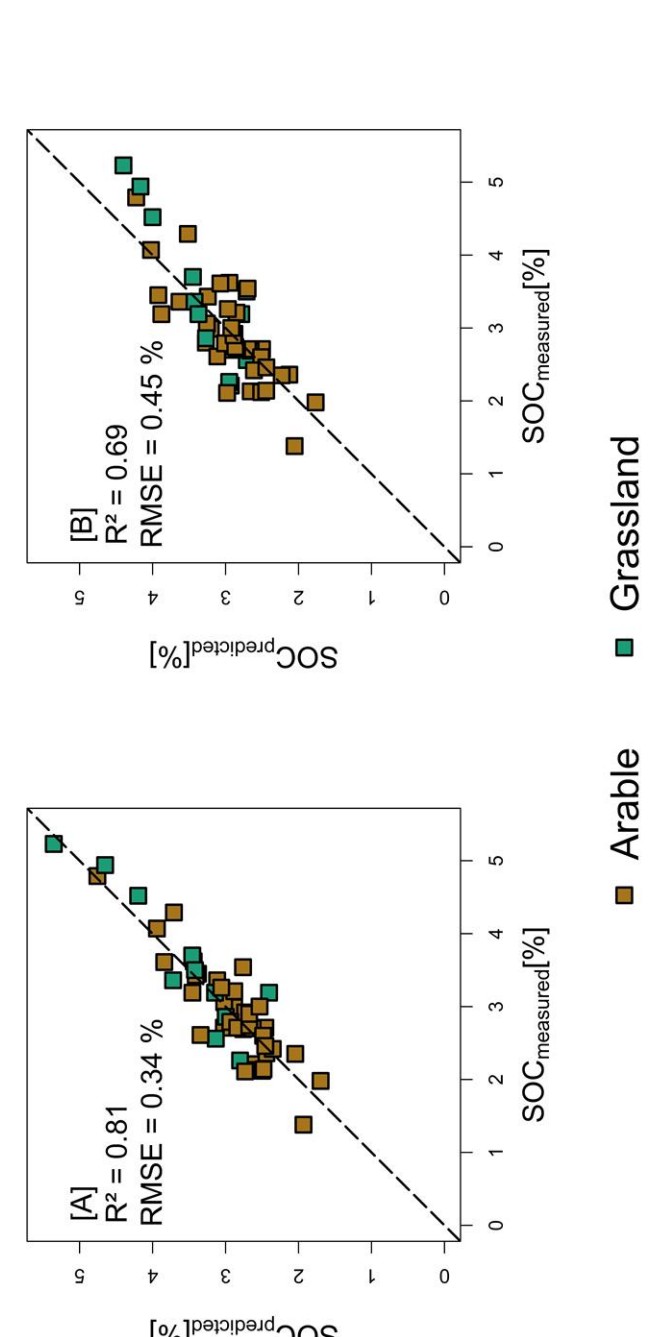

**Fig. 6.** Predicted *vs.* measured content of SOC of soil samples from the DCS cluster; predictions based on the local model [A] and the global model [B].

## 4 Discussion

### 4.1 Bivariate relation of soil mineral phase characteristics to SOC and its labile fractions

Our study showed that interactions of SOC with the mineral phase are highly relevant for the content of SOC as well as of its labile fractions HWEC and MBC in soils. High regression coefficients of SOC to fSilt plus clay (Table 3) agree with reports on the relevance of organo-mineral associations for the stabilization of SOC and related to this the accumulation of the labile fraction HWEC and MBC (Lützow et al., 2006). Furthermore, sandy soils contained the lowest content of SOC while loamy and silty soils had an equally higher content of SOC (Table 1). This is typically expected and confirms numerous previous reports, e.g. Ludwig et al. (2003) and Vos et al. (2018). In contrast, for the LBS cluster with its very sandy soils a slightly positive effect of sand on SOC was found. This, however, is most likely a consequence of agricultural practice, with high manure application to the LBS soils in the sampled area. This was further confirmed by a factor of 1.2 higher ratios of SOC/N and HWEC/N as well as by a lower oxygen content of SOM compared to soils of the other parent material clusters (factor of 0.6; Table 1). Besides parameters directly related to soil texture, pedogenic Al- and Fe-oxides were found to be strong predictors of SOC in soils. For example, Kaiser and Guggenberger (2000) and Lützow et al. (2006) showed that Al- and Fe-oxides have a relevant influence on the accumulation and stabilization of SOC as well as a high affinity to retain components of the labile SOC fractions (Kaiser and Zech, 1998; Kaiser et al., 2002). Although soil acidity strongly affects soil processes such as microbial activity and turnover that are relevant for SOC accumulation (Kemmitt et al., 2006), no clear relation between pH and SOC or its labile fractions was found by bivariate linear regression. Yet, soil parameters that are strongly related to soil acidity, i.e. ECEC as well as the content of exchangeable polyvalent cations such as $Ca^{2+}$ and $Mg^{2+}$, were suitable predictors for SOC and its labile fractions in this and previous studies (O'Brien et al., 2015; Rasmussen et al., 2018). This is causally explained by the stabilization of SOC in organo-mineral associations and the contribution of multivalent cation bridges ($Ca^{2+}$ and $Mg^{2+}$) to it (Kaiser et al., 2012). The even higher ability of the content of pedogenic oxides to explain variance of SOC and its labile fractions was indicated in this study for several clusters (total and local) by bivariate regressions (Table 3). This corresponds to findings of Rasmussen et al. (2018). They found a prevalence of pedogenic oxides in humid areas with moderately acidic soils, while exchangeable Ca and clay prevailed in soils of dry climates with circumneutral to alkaline pH. Such a case-specific prevalence of parameters to predict SOC, HWEC or MBC demonstrates that it is preferred to use specific parameter sets when it is aimed to focus on local areas. In this study ECEC and $(Ca+Mg)_{ECEC}$ were not generally applicable as predictors but it strongly depended on the parent material and texture cluster. For example, ECEC and $(Ca+Mg)_{ECEC}$ were found to be relevant for the clusters of DLS and PSS, while for DCS they were of minor importance. The bivariate models revealed that the stone content had only a small impact on SOC, HWEC and MBC. Hence, a funnel effect of the stone content, by funneling more SOC into the remaining fine textured soil (Bornemann et al., 2011) was irrelevant.

### 4.2 Explanation of variance by multivariate parameter combinations for local and total datasets

The combinations of factors and soil properties affecting SOC and SOC fractions, respectively, were dissimilar between the different local areas investigated in this study. The PCA revealed that differences according to parent material and soil texture were most relevant to separate the dataset into various local clusters based on different factors (Fig. 2 A and B; Table 2). At the same time, this illustrates the importance of the mineral composition (parent material) and grain size (soil texture) for the accumulation of SOC as well as its labile fractions HWEC and MBC. In contrast, land use was not useful for a separation into clusters. This was unexpected because typically topsoils under grassland have higher SOC contents compared to arable soils (Poeplau et al., 2020), which was largely confirmed for the samples investigated in this study (Table 1). This went along with

differences in the composition of SOM (Table 1 and Table S1). However, data ranges of SOC, HWEC and MBC contents were largely overlapping and similarities even increased in PCA when further soil properties were included. In comparison, mineral phase soil properties clearly separated the dataset. Consequently, a broad scatter of the land use clusters was obtained by PCA, suggesting to treat the land use clusters as total datasets as well.

Several studies with large datasets covering national or continental scales, e.g. soil inventories, pointed out the relevance of combinations of multiple factors and parameters instead of using single predictors to estimate SOC or its labile fractions (Wieder et al., 2015; Vos et al., 2018; Gray et al., 2019). Furthermore, local studies covering small areas with narrow ranges of soil properties often show weak bivariate relationships between SOC and components of the mineral phase or environmental factors (Jian-Bing et al., 2006; Liddle et al., 2020). Accordingly, models focused on specific local clusters and combined with multiple parameter sets were superior compared to the total model that was developed for the total (entire) dataset to estimate SOC, HWEC or MBC (Fig. 6). The different parameter combinations indicate that distinct properties of the mineral phase control SOC, HWEC and MBC in the soils of the different clusters. Understanding SOC as a continuum (Lehmann and Kleber, 2015) implies that accumulation of SOC is a multidimensional process, where SOC is simultaneously affected by several soil properties and factors. Hence, SOC accumulation and variability depends on various interacting factors and soil properties, respectively. The substantially lower ability of bivariate models to estimate SOC compared to multiple parameter models is in line with this assumption. Accordingly, it was superior to use multiparameter mixed effect models to estimate SOC and the two labile fractions. Especially parameter combinations within the land use clusters gained a high-explained variance (Table 3, Table 4). A comparison with studies on regional or national scale (Vos et al., 2018; Mayer et al., 2019) suggests that the importance of factors such as land use, soil texture or parent material varies with the observed scale. Wiesmeier et al. (2019) reported that soil texture, land use and land management are relevant to explain SOC variability at all scales. On regional or larger scale, also environmental factors such as climate, geology, soil use, topography are relevant for SOC. Yet, at a local or smaller scale factors such as climate become less important, while parameters representing small-scale soil physico-chemical properties gain importance for explaining the variability of SOC. Thereby, different factor and parameter combinations were identified for the different local clusters by mixed effect modelling. The prevalence of a parameter for quantification of SOC can differ dependent on environmental factors (Rasmussen et al., 2018).

Consequently, the quality of the multiparameter models was further improved by the implementation of local specific random effects such as parent material or land use. Dependent on the random factors parent material, soil texture class and land use different parameter combinations explained SOC, HWEC or MBC (Fig. 4 and Fig. 5). For the total dataset, nearly all predictor parameters showed a significant contribution to the explanation of SOC. Most of these soil mineral phase parameters were also significant in linear regression. In contrast to the bivariate models, most mixed effect models revealed parameters related to soil acidity as significantly important to estimate SOC, HWEC and MBC. This highlights the importance of soil acidity on SOC dynamics due to its effects on the reactivity of the mineral phase and the activity of microorganisms (Hillel, 2004). In order to explain the variability of HWEC and MBC for the various local clusters, different combinations of mineral phase parameters were required that also clearly differed from the parameter combinations used in the models for SOC (Fig. 4 and Fig. 5). Such differences concerning significantly contributing parameters were also found by other studies for specific clusters or local sampling sites (Heinze et al., 2018; Quesada et al., 2020). This emphasizes that local models are required and superior when it is the aim to estimate SOC and SOC fractions on a local scale. The total models used for the total datasets in this study reached the best predictions for SOC, HWEC and MBC. Yet, this was largely biased by the large samples size; applying the same total models to local samples sets produced clearly poorer estimates compared to the more specific local models as indicated by the explained variance and the RMSE (Fig. 6; Table 5 and Table 6). Consequently, aggregation of smaller datasets, e.g. from a local

scale, to a larger dataset enables to predict SOC and its labile fractions to a higher extent. In opposite a model that was derived from a total dataset and is applied to a local dataset with smaller ranges of properties is less suitable, resulting in a variance explained on a lower level. Depending on the properties of the soil mineral phase, each specific cluster was controlled by other properties, which best explain the accumulation of SOC and its labile fractions. This implies the importance for analysis of local clusters to avoid a subordination by models of total datasets.

4.3 Distinct model parameters for the estimation of SOC, HWEC and MBC Comparing the results of mixed effect models using the different random effects (parent material, soil texture, land use), the models using parent material yielded highest explained variance for the estimation of SOC. For HWEC and MBC best predictions at a high level of explained variance were obtained by models using land use as random effect (Table 4). High explained variance mostly went along with smaller RMSE values. The parent material predefines the boundaries for accumulation and stabilization of organic matter (Gray et al., 2015). The importance of land use as random effect especially for the labile fractions results from the fact that these are especially influenced by soil management (Cardoso et al., 2013; Lal, 2016). In general, the variance explained by the mixed effect models was not similar, but varied between SOC and its labile fractions HWEC and MBC. It became clear that SOC and the labile fractions HWEC and MBC are not fully correlated but quantitatively quite distinct SOM pools with different dynamics (Wander, 2004; Tokarski et al., 2020). Not last it is suggested that the faster turnover of the labile fractions is one of the reasons for the lower explained variability by the different models. HWEC is a measure of bioavailable and degradable organic carbon (Weigel et al., 1998). Although it is closely correlated to SOC ($R^2$=0.75) it is best estimated by distinct parameter combinations compared to SOC, which is explained by the substantially higher variability of HWEC (Table 3 and 4). Changes in HWEC are mostly assigned to inputs of organic fertilizer substrates (Weigel et al., 1998) and the soil management (Ghani et al., 2003). For MBC especially soil management and factors such as C-input, climate, soil texture and soil pH are relevant (Wardle, 1992). Accordingly, the effect of land use but also of soil texture was most relevant for MBC accumulation. Similar to findings of Ludwig et al. (2015), MBC increased with the content of silt and clay but declined with sand, which is explained amongst other by the contribution of MBC to aggregate formation, the habitable surface and accessibility of SOC (Totsche et al., 2018). Additionally, management practices such as tillage and the application of organic fertilizer directly influence MBC (Liang et al., 1997). Lower explained variance of HWEC and MBC compared to SOC were based on a smaller relevance of the mineral phase parameters for their accumulation. Labile fractions such as HWEC and MBC, containing larger proportions of bioavailable and easily degradable organic compounds, are subject to faster turnover (Landgraf et al., 2006; Lorenz et al., 2021).

## 5 Conclusions

The reliable estimation of SOC and of its labile fractions HWEC and MBC is a task of growing importance in order to manage soil properties and functioning. That task will most often focus on local soilscapes with minor variation range in soil properties. This study showed that local models are superior to total models. Mixed effect models combined with random effects yielded best estimates and highest explained variance for SOC and even its labile and quite dynamic fractions HWEC and MBC. For this purpose, the application of multivariate approaches to estimate SOC, HWEC and MBC clearly resulted in a higher explained variance compared to models based on bivariate linear regressions. Even a reduced dataset, representing parameters of the soil mineral phase is suited to estimate contents of SOC as well as HWEC and MBC. The inclusion of overall factors such as parent material, soil texture class and land use as random effects further improves the models. Total or even global models, developed from large-scale studies across countries or continents, often reach best estimates; however, they are subordinate for the above-mentioned small-scale areas and low sample numbers. Application of total models to local datasets leads to a smaller explained

variance and higher RMSE. From a practical perspective, the selected set of soil mineral phase parameters can be easily determined by using well-established methods and the parameters are rather stable over a longer-term. Thus, using such parameters for the estimation of SOC, HWEC and MBC is expedient. The presented research will be further enlarged by studying larger datasets containing more clusters in order to better identify local drivers of SOC and of its labile fractions. Our research shows that local models, considering site-specific parameter combinations, are superior to total models, although they are based on much smaller datasets. If such local datasets and models are available, they should be preferred. For further research we suggest to assess even larger datasets, in order to find out whether local subclusters can be identified and to examine if these clusters are best explained by total or local models. Furthermore, research is needed to determine most relevant parameters for a site adapted estimation of SOC and its labile fractions on different landscapes.

## 6 Code/Data availability

The raw data is available upon request to the authors.

## 7. Author contribution

MO, TU, MV, STB conceived, and designed the study. MO, MS, SS performed the sampling and analysis. MO wrote the first draft. All authors (MO, MS, SS; UT, MV, STB) contributed to generating and reviewing the subsequent versions of the manuscript.

## 8. Competing interests

The authors declare that there is no conflict of interest.

## 9 Acknowledgements

This study was founded by the German Environment Agency in the framework of the ScreeSOM project ("Screening methods for a cost effective detection of supply with SOM in arable and grassland soils", project no. 371 673 208 0). The authors thank Marc Marx for collegial cooperation. We are grateful to the colleagues of the Soil Science Department of Trier University, P. Ziegler and E. Sieberger and the students A. Forens, M. Heinrich, K. Struwe and A. Hergert for assistance during field and laboratory work. Not least we thank two anonymous reviewers for their valuable comments.

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
