# Peer review of "Content of soil organic carbon and labile fractions depend on local combinations of mineral phase characteristics"

_SOIL, 2021_

## Author Comment (AC1)

**Authors response to Reviewer 1 (RC 1) for soil-2021-81**

In their manuscript, 'Soil organic matter and labile fractions depend on specific local parameter combinations', Ortner et al. present their work on the analysis of factors controlling the soil organic carbon (SOC) concentration in topsoils of the region around Trier, Germany. The authors collected topsoil samples in arable land and grassland in 4 regions with different parent material, and determined the organic carbon (OC) concentration, hot water extractable carbon (HWEC), microbial biomass carbon (MBC) and multiple soil properties on these samples. They used PCA to cluster the soil samples based on parent material and soil texture into different clusters. The aim of their study was to assess the main factors controlling topsoil organic carbon concentration, HWEC and MBC using two modelling approaches: a bivariate model and mixed effects models. The main findings are that (i) mixed effect models outperformed bivariate (linear) models in predicting OC%, HWEC and MBC, (ii) at the local scale, site-specific parameters explained OC variability better than landscape-related variables and (iii) using the 'local' model resulted in better results when predicting the OC% of a specific cluster compared to the 'global' model.

The results of the present study help to improve our understanding of the factors controlling topsoil organic carbon concentrations at the landscape scale, which is needed e.g. in order to improve soil organic carbon models. The authors have constructed a valuable dataset which may benefit other researchers. I would therefore encourage the authors to make this data available through an online repository, instead of making it only available upon request.

We thank for this comment. The dataset was generated in the framework of a contract project of the UBA. We aim to clarify with UBA if the data can be fully published.

Overall, the manuscript is well-written. However, at multiple locations very long sentences are used, which does not benefit a smooth reading. Splitting those sentences and using more commas would improve the readability of the manuscript considerably. In addition, I would encourage the authors to use subsections in the Results and Discussion sections, which will provide a better overview to the reader of what is being presented and discussed.

Following the advice, we revised and split several of the longer sentences. Further, we took up the valuable hints regarding subsections for the results and discussion sections.

One of my main concerns about the present manuscript is related to the quantification of the goodness-of-fit of the different models, which is now done using R-square. This is a measure to quantify the proportion of variation in a dependent variable that is explained by an independent variable, but is not a measure for the goodness-of-fit of a model. For example, a very poor model can have a high R-square value, while a good model can have a relatively low R-square value. Therefore, the authors should use a different measure to quantify the goodness-of-fit of their model when comparing measured with modelled data, such as the (root) mean square error or similar.
We agree that R2 is good to show the percentage of explained variance but not fully sufficient to document the goodness-of-fit of a multivariate and/or non-linear

model. Hence, the RMSE was added as a measure for the goodness-of-fit. The presentation of R2 was reduced to the bivariate linear models.

In addition, I missed a discussion about the broader implications of the results and the implications for future research. For example, do the authors suggest that researchers should use 'local' model whenever possible? And how about regions where local information is not present? It would also be very informative if the authors would quantify the difference in predicted SOC% when using a global versus local model. To how much of an over or underestimation would this lead? Is that difference significant enough to invest more resources in the collection of local data?

Statements about broader implications and some recommendations for further research were added to the conclusions.
The level of over- or underestimation is represented by the RMSE. We added the RMSE

Another concern is related to the title, which I find not very informative. For example, it will not be clear to someone who has not read the manuscript what 'specific local parameter combinations' are. Also, it would be good to be more specific about what they mean with 'soil organic matter and labile fractions'. From the title it is not clear if the authors mean SOC concentrations, stock, spatial distribution etc. In addition, the manuscript discusses soil organic carbon, and not all soil organic matter, so would be good to change this in the title.

We changed the title in order to make it clearer: "Content of soil organic carbon and labile fractions depend on local combinations of mineral phase parameter"

Lastly, it would good if the authors specify in the beginning of the manuscript that they discuss SOC concentrations, and not stocks. Throughout the manuscript, the authors talk about 'SOC' without specifying that it concerns concentrations, not stocks. This is an important difference, which should be clear to the reader from the abstract onwards, and repeated throughout the manuscript. For example, the authors could change 'SOC' to 'SOC%' to make this clear.

In the text (e.g. Abstract) and title it is now explicitly mentioned that concentrations were investigated.

**Answers to specific comments RC1**

L18: Would be good to explain here what you mean with 'global' and 'local' clusters (and models).

Thank you for this hint, we added the definition for the investigated clusters. Further we decided to replace the term 'global' by 'total' to prevent any confusion regarding different scales (local vs. global scale). It should be now clearer that we talk about the total dataset encompassing the different local datasets.

L19: define that you assess SOC concentrations, and thus not stocks

See general comments. We added this information to the title, Abstract as well as in the Introduction and in Material and Methods. In Tables and Figures SOC is given in % as unit, indicating that we assessed concentrations.

L20: would be good to explain here which 'labile fractions' you study

Information regarding the labile fractions was added as requested. "… explaining SOC and its labile fractions hot water-extractable C (HWEC) and microbial biomass C (MBC)".

L27: here you use the term 'organic matter', while until here you used 'SOC'. Please be consistent with these terms, and only use one
It was changed to 'organic carbon'.

L29-30: 'showing that labile fractions depend less on soil properties than on organic matter input and turnover in soil'. The latter were not studied, so you cannot say this with certainty. Would be better to end the abstract with a statement about the broader implications of your results.
We thank for that comment. To avoid the impression that organic matter input and turnover were investigated in this study, we changed the sentence: "showing that labile fractions depend less on soil properties but presumably more on processes such as organic carbon input and turnover in soil."

L41: another important labile fraction of SOC is particulate organic carbon. Would be good to justify why you did not study this fraction
The authors fully agree that particulate organic carbon (POC) is an important labile fraction. Due to trivial financial reasons we had to decide which fraction(s) we can study. We decided for hot water extractable carbon (HWEC) and microbial biomass carbon (MBC) because they are methodically clearly defined. Quite often it is stated in the literature that both are very closely correlated with each other, and thus deliver no different information. We hypothesized and aimed to show that this is not the case (which was confirmed in this study). Additionally, we decided against POC because it is not uniformly defined, either by size or by density. So we hope that HWEC and MBC are representative measures of labile SOC pools. Again, we fully agree that having additional data on POC would have been a perfect completion of the dataset.

L45: 'MBC is expedient to explain SOC dynamics': this is rather vague, please be more specific
To make the point more clear, we added the following sentence: "Additionally, labile carbon fractions such as MBC quantitatively dominate in short-term turnover processes, while changes in SOC will only become significant over periods of decades. Therefore, MBC is expedient to explain SOC dynamics".

L45-46: 'much less research and attempts for quantitative modelling of these labile fractions […]': recently, multiple mechanistic models have been used to simulate labile carbon fraction such as MBC and POC, e.g. Ahrens et al. (2015), Wieder et al. (2015) and Zhang et al. (2021)
Thanks for this valuable comment. We changed the sentence and included some recent publications on modelling. We left the statement that SOC is mostly considered for such simulations, while there is still a need to take labile fractions more into account in order to gain a better understanding of SOC dynamics.

L58: 'In addition or even instead of': choose one
Ok, done.

L63: please clarify what you mean with parameters versus factors, as you use these terms throughout the manuscript
We added some examples. In general parameters include soil properties on a interval or ratio level of measurement while factors were applied on a nominal level of measurement.

L87: please define what you mean with 'global models'

To avoid confusion, we replaced both terms 'global dataset' and 'global model' with 'total dataset' and 'total model'. The total model is based on all data of the total dataset that encompasses all local datasets.

L99: the term 'sufficient quantification' is rather vague, please clarify this
The sentence was changed as follows: "It was aimed to determine the suitability of local models in comparison to total models to achieve an improved quantification of SOC, HWEC and MBC for local landscapes with distinct properties.".

L107: 'similar numbers of samples': how many per region?
Number of samples taken per region were shown in Table 1. Additionally, they were now added to the sentence in brackets for each sampling region.

L108-109: the use of the abbreviations throughout the manuscript is not intuitive and confusing for the reader, please use different names to identify the different regions, e.g. the parent material
We agree that abbreviations are a compromise between clarity and readability. Using the full terms or terms such as 'Muschelkalk' and 'Luxemburg sandstone' would have been too long, though. Shorter abbreviations were also inconclusive, e.g. schist and sandstone are both abbreviated 'S'. Hence, we plead for keeping the chosen abbreviations.

L119: why were some samples stored at -20 °C and others air-dried?
Samples were stored until they were analyzed. Storage was done in a uniform way for all samples. One part of each sample was air dried for subsequent chemical and physical soil analysis, another part was kept moist and was frozen for subsequent soil microbial analyses (MBC, MBN or respiration). This is now clarified in the text.

L134: was the chloroform fumigation extraction performed on samples freshly collected from the field?
Chloroform fumigation extraction was done on sieved material that was stored at -20°C before analysis. This was done to avoid changes until measurement was conducted. The suitability of this storage was proven in preliminary projects (data not shown).

L137-140: for how long were the samples incubated? How often was the CO2 measured?
Samples were preincubated at room temperature for one week (7 days), measurement was conducted for 24 hours at an interval of one hour. The information was added to the text.

L143: were all parameters log-transformed? Please clarify this
To conduct the principal component analysis all variables were log transformed to receive standardized and comparable variables. The information is contained in the text.

L146: please provide some examples of the 'mineral phase parameters'
We added two examples ($Fe_o$ and fSilt+clay) into this sentence.

L146-147: please provide more information about the linear regressions that were performed

We added the information that we applied linear regressions using single predictors, and information that we checked the residuals for normality.

L156-157: Please provide information about which parameters were removed from the models
The non-significant parameter with the highest p-value was removed from the model. This was repeated until all remaining parameters were significantly contributing to SOC, HWEC or MBC. This information was added to the sentence.

L161: were all parameters checked for collinearity? Please clarify
We checked all mineral phase parameters for collinearity, which were used by the mixed effect models. Based on this, it was found that soil texture components (Sand, c+mSilt and fSilt+clay) showed collinearity as well as ECEC and Ca+Mg$_{ECEC.}$ We clarified this in the text.

L163: why a square root transformation? Please justify this
Square root transformation was selected as a common transformation and was suited to achieve normal distribution and heteroscedasticity of the residuals.

L163: Please clarify how the performance of the models was examined
Basically, we started by comparing the explained variance and, based on your valuable comments, now also added RMSE as indicator for performance.

L170: Please clarify the difference between 'soil' and 'topsoil' properties
Topsoil was separately mentioned due to the fact that our study focusses on agricultural topsoils. To avoid confusion or misunderstanding we decided to use only the term 'Soil properties'.

L177: are those differences statistically significant? What are the averages for the different parent materials?
There are some statistically significant differences, averages for the parent materials are given in Table 1 as mean ± sd.

L186: please provide examples for the 'parameters describing the composition of SOM'
We now mention some examples in the text, such as SOC, Nitrogen, hydrogen or oxygen, HWEC or MBC.

L191-194: this is not clear
We rephrased these sentences to make it clearer. Clusters identified by the PCA cover a different number of samples of the total dataset. Based on this clusters including the vast majority of samples were considered to represent the total dataset, while substantially distinct clusters, including only a part of all samples, were considered to represent local datasets.

L205-206: which 10 parameters?
Selected parameters were shown in Table 1 and in Table 3. Further we mention examples of these parameters in the Material and methods section. Examples of these parameters were added to the text in brackets.

L213: 'that largely matched with those found for the complete dataset': this is not clear
We adapted this sentence to make it clearer.

L224: what do you mean with 'sufficient extent'? Similar wording is used throughout the manuscript, but this is very subjective and should be clarified.
Thanks for this hint, we checked the manuscript and exchanged such phrasings by objective formulations using statistical parameters is applicable. See also L369

L240: please clarify what 'equal weight of samples' means
It means that both clusters (arable and grassland) contain a similar number of samples from each parent material resulting in a broad range for each soil property, catching up the properties from soils of each sampling region. We rephrased the sentence to clarify its meaning.

L237-242: please make clear that you are discussing the results of the bivariate models
We now mention that these lines address the bivariate regressions.

L241: what are the 'complex interactions of several different parameters'?
The term 'complex' was deleted. It makes sense concerning the environmental interaction of these parameters but not concerning the contribution to a mathematical model.

L243: please clarify what you mean with 'insufficient'. Which measure do you use to determine if a model performance is sufficient or not?
The authors thank for this hint, we changed such phrasings to objective formulations.

L249-251: R-square values are no measure for model performance, please provide the root mean square error (or a similar measure). Please show these results in a graph, perhaps in the supplement?
R-square is used to show the explained variance, this manuscript aims to show how much mineral phase parameters and their different combinations are able to explain the variance of SOC, HWEC and MBC. Notwithstanding, we fully agree that the root mean square error is a much better measure to determine the model performance. Therefore we added it to the text.

L273: do you mean the bivariate models with 'linear regressions' Please be consistent with this terminology
Yes, it means the bivariate models, we made it clearer.

L284: please replace R-square with a measure of model performance
See the above response. Further, as a measure for mixed effect models we added marginal and conditional $R^2$ to describe the $R^2$ directly related to these models.
$R^2$ based on predictions is only able to give a pseudo $R^2$ which is based on a linear regression between predicted vs measured. Such comparison between predicted vs measured and the received pseudo $R^2$ was technically the only option to test the performance of a total model, when applied on a local dataset. This information was added to the text.

L287: please provide the goodness-of-fit values before concluding that a certain model has an 'inferior performance'
We added this information, but we also kept the $R^2$ because it was aimed to investigate which model explained the variance to the highest extent.

L309-310: by saying 'Al- and Fe-oxides were shown to have a relevant influence on sequestration and stabilization of SOC', it seems like you explicitly studied this, while you only used a statistical model to assess this. Also, since you model SOC concentrations and not stocks, you cannot say anything about C sequestration, as this also depends on bulk density.

This sentence was linked to a reference and started with the term 'accordingly' in order to emphasize that this mechanistic interpretation of our statistical finding is not based on our study. We deleted the term 'sequestration' as requested since we do not address SOC stocks.

L342: please explain what you mean by 'multidimensional'

Multidimensional means that SOC is simultaneously affected by serval soil properties and factors which explain the overall accumulation and variability instead of single one to one interactions.

L348: 'to explain SOC': please clarify which aspect of SOC

The factors mentioned in the sentence were able to explain SOC under different scales and environmental conditions but in general, these factors enable to explain the accumulation and the variability of SOC.

L366: how do you conclude that sample size biased the results? Did you test for this?

It is true, randomly selecting a data subset from a total dataset does not necessarily lead to different (biased) modelling results. However, in this study total clusters including a larger number of samples showed a higher explained variance, which is a consequence of a broader variety of soil properties in the dataset. Local clusters with a smaller sample size also showed smaller ranges of the tested soil properties, leading to models with a lower explained variance.

L369: 'satisfying extent': how is it quantified that a model performs satisfying? Please be objective in deciding if a model is good or not

Thanks for this hint, we checked the manuscript and exchanged such phrasings by objective wordings.

L370: what do you mean with 'partially practicable'?

We replaced the term with 'less suitable', which is based on the lower statistical performance.

L374-375: 'sufficient quality level': same remark as L369

Similar to comment to L369, we changed the wording.

L380-381: by saying 'It became clear that […] with different annual dynamics', it seems like you tested annual dynamics. Please rephrase

We rephrased this sentence accordingly.

L281-382: You did not take SOC turnover into account, so how can you say that this is a reason for the lower explained variability by the different models?

An aim of this study was to investigate the linkage between mineral phase properties and labile fractions. Compared to SOC we found a lower explained variance for the labile fractions. Hence, we assume that – although we didn't explicitly investigate it - the known faster turnover of these fractions (depending, e.g. on land use management) will significantly contribute to the concentration of HWEC and SOC, thus explaining the gap in explained variance of HWEC and MBC.

L403-404: 'sufficient estimation': same remark as L369
Similar to comment to L369, we exchanged this formulation.

L405: would be good to end the Conclusions section with a statement about the broaderimplications of your results
We added the following statement: "Our research shows that local models, respecting site-specific parameter combinations, are superior to total models, although they are based on much smaller datasets. If available, they should be preferred."

**Figures and tables**
Fig. 2: the colours in B are difficult to distinguish
It was changed accordingly.

Table 3: please make clear in the caption that these are the result for the bivariate Regressions
It was changed accordingly.
Fig. 3: 'Predicted vs. measured': please clarify in the caption which model was used to make these predictions. Please provide a measure for the goodness of fit and remove the R-square values, as this is not measure for model performance
We now mention it in the caption and added RMSE as measure for model performance.

Table 5: please provide more information about the table in the caption, the table should be clear to the reader without having read the entire manuscript. It would be more informative to provide a table with e.g. root mean square errors instead of R-square
We added some information regarding the RMSE, but we also want to show how the models differ in their explained variance. So we kept $R^2$.

Figure 6: it would be informative to show the same graphs for other clusters in the supplement. Please remove the R-square values and replace them by a measure for the goodness-of-fit of the models
Fig. 6 shows the performance of the previously developed total model, when applied to a local dataset. The model was not fitted to the data of the local dataset (which would have yielded the local model). Consequently, pseudo $R^2$ is given as a measure to compare the agreement (or disagreement) of modelled vs. measured data. Additional, we also added the RMSE to this Figure.

**Technical comments**
L36: driver => drivers
Done

L57: expedient => suitable
Changed

L73: space between 'asCa2+'
Changed

L119: it's not clear what 'respectively' refers to
Removed due to changes in this sentence.

L170: it's not clear what 'respectively' refers to
Ok, rephrased

L172: it's not clear what 'thereby' and 'essentially' refer to
Ok, rephrased

L252: it's not clear what 'respectively' refers to
Ok, rephrased

L275: what do you mean with 'not last'?
We want to highlight that soil acidity and its describing parameter were also relevant. The typo, however, was corrected to 'not least'.
L304: 'the in total very sandy soils': please rephrase
Ok, rephrased
L320: what is 'circumneutral'?
Circumneutral means soil pH that is close to neutral or neutral having a pH between 6.5 and 7.5. It is an established term. See for example:
Carl O. Moses, Janet S. Herman, 1991, Pyrite oxidation at circumneutral pH, Geochimica et Cosmochimica Acta 55/2, 471-482.

L324: please remove 'respectively'
Done.

L343: please remove 'respectively'
Done.
L344: confirmed => is in line with
Done.

References
*These references were chosen based on their scientific content. I leave it up to the authors to decide if they wish to include them in their manuscript.*
Thanks for this valuable references, we added some of them to our manuscript.

Ahrens, B., Braakhekke, M.C., Guggenberger, G., Schrumpf, M., Reichstein, M., 2015.Contribution of sorption, DOC transport and microbial interactions to the 14C age of a soil organic carbon profile: Insights from a calibrated process model. Soil Biology and Biochemistry 88, 390–402. https://doi.org/10.1016/j.soilbio.2015.06.008

Wieder, W.R., Grandy, A.S., Kallenbach, C.M., Taylor, P.G., Bonan, G.B., 2015. Representing life in the Earth system with soil microbial functional traits in the MIMICS model. Geoscientific Model Development 8, 1789–1808. https://doi.org/10.5194/gmd-8-1789-2015

Zhang, Y., Lavallee, J.M., Robertson, A.D., Even, R., Ogle, S.M., Paustian, K., Cotrufo, M.F., 2021. Simulating measurable ecosystem carbon and nitrogen dynamics with the mechanistically defined MEMS 2.0 model. Biogeosciences 18, 3147–3171. https://doi.org/10.5194/bg-18-3147-2021

---

## Author Comment (AC2)

**Authors response to Reviewer #2 (RC 2) for soil-2021-81**

This manuscript presents the distribution of SOC and its labile fractions predicted using parent material, land use and soil properties in Southwest Germany. The results indicated that soil properties were clustered by parent materials and soil texture rather than land use. In general, mixed-effect model gave better predictions than bivariate regression. They compared "global model" with "local model" to show that the application of global model on local dataset introduced poorer predictions. Also, the explained variance generally decreased from bulk SOC to its labile fractions.

In general, the objectives were clear and relevant while the scientific value is sufficient. The large sample size contributes to a robust prediction. However, there are several concerns to be addressed.

One concern is the distribution of the sampling points. As mentioned in L47-48, soil formation is also controlled by climate and topography. The clustered locations of the four parent materials are likely to introduce differences in topographical and climate conditions. As climate and topography factors were not included in the models, their effects might be recognized as the effects of parent materials, texture or land use in the predictive models. (Details in comments for Fig. 1)

See answer to comment on Fig. 1

Another concern is that the usage of "global/local scale", "global/local model", "global/local cluster" and "global/local/entire dataset" may confuse readers because they were used without necessary explanations. In addition, the words "global" vs. "local" give the impression that the study aimed to compare SOC distribution on global vs. local scale, but no investigation on global scale was given in this study.

Thanks for your comments. We fully agree that the term 'global' may be confusing. Therefore, we decided to replace 'global' by 'total' to avoid this misunderstanding.

In addition, in some parts of the manuscript, R^2 was used to estimate whether models are well-fitted, which is not proper. Also, the Results and Discussion can be improved by splitting them into sub-sections and better re-organizing. Finally, the readability of the manuscript can be improved by revising long-complexed sentences and vague expressions.
As similar mentioned to Reviewer #1, R-square is used to show the explained variance, this manuscript aims to show how much mineral phase parameters and their different combinations are able to explain the variance of SOC, HWEC and MBC. Notwithstanding, we fully agree that the root mean square error is a much better measure to determine the model performance. Therefore we added it to the text.
Title: (1) Although "soil organic matter" is used in the title, the main part of this manuscript is mostly talking about "soil organic carbon". Please be consistent in using them because soil organic matter contains not only organic carbon but also other elements such as nitrogen.
We agree that a consequent use of the terminology is required. Therefore we changed the title to "soil organic carbon"

(2) It is advised to add restrictions on the area/location because the study was performed in western Germany and will not be necessarily applicable in other places.

We agree that local areas were sampled that were all located in the larger region of Trier in southwestern Germany. Anyhow, it was not our primary intention to characterize a specific region. Instead, the sampling region was selected because it covers soils with identical land use types (arable and grassland) and similar climatic/pedoclimatic conditions but substantially different parent material, and thus different soil mineral phase properties. Our prior aim was to show that SOC of local clusters is better assessed using local models. It is clear that at larger scales (nation-wide and larger) differences in pedoclimate add to the factors explaining SOC (and labile fractions). However, the climatic impact is generally not relevant for local areas independent of where they are. At the same time what the reviewer states is exactly would we found and suggest: A specific assessment of local areas with a local model is preferred. Such a local model is not transferable on to one to another local area.

L14, L18 and L21: It is confusing to mention "local scale", "global/local cluster" and "global/local dataset" in abstract without further explanation. The usage of "local" vs."global" gives me a feeling that this study compares SOC distribution on local vs. global scales. Apparently, the distribution of the sampling sites represents a local or sub-regional scale. It is suggested to either give them definitions when they are mentioned for the first time or replace them with more suitable words.

Thanks for your comment, we now define what is meant by local and total (previously global) cluster/dataset and we replaced the scale-related terminology.

L21: As only regressions were performed in this study, it is recommended not to use both correlation and regression in the text.

We revised the text accordingly.

L21-23: It is difficult to understand this sentence. It is not clear between which factors the correlations are significant. What does "partially low" mean? Splitting this sentence into simple ones may help.

For a better understanding of the sentence, we followed the advice to split it. 'Partially low' means that some of the correlation coefficients ($R^2$) only showed a small explained variance. Such vague terminology was replaced by more objective wordings.

L66: In general, organo-mineral associations are considered contributing to the formation of stabilized fractions (not labile fractions) and therefore the accumulation of SOC.

We agree that the formation of organo-mineral associations leads to the stabilization of SOC. Additionally, such accumulation of SOC goes along with increasing contents (not stabilization) of labile fractions such as DOC that are only weakly retained through other mechanisms in the presence of pedogenic oxides. This is what we wanted to say. We changed the sentence as follows to make it more clearer: "Organo-mineral associations are highly relevant for stabilization and accumulation of SOC and also for the accumulation of its labile fractions (Lützow et al., 2006)."

L72: …leading to SOC sequestration…
We adapted this line.

L85: Please check if surnames and given names are misplaced in this reference.
Checked, there is no misplacement of surname and given names. Surname is Jian-Bing, given name is Wei. See https://link.springer.com/article/10.1007/s10661-005-9158-5#article-info

L90: "Local vs. global models" are confusing. Do they mean models on local vs. global scales?
We wanted to indicate that it is necessary to apply models on local clusters/datasets instead of on one global (total) dataset to best explain SOC (more precisely its variance).

L102: Is the "entire dataset" equivalent to the "global dataset"?

A global (now total) dataset is defined as a dataset encompassing the large majority of the dataset. Therefore, next to the entire dataset, the clusters of arable, grassland and loam act as total (previously global) dataset.

Materials and Methods

L104: It is recommended to add more information about the study area. In general, most studies show readers climate factors (e.g. annual precipitation and average temperature), soil type/classification and composition of vegetation/crops.

We added additional information regarding the study area.

L119: Please explain why soil samples were stored either at -20 °C or air-dried. For different analyses?

Samples were stored until they were analyzed. Storage was done in a uniform way for all samples. One part of each sample was air dried for subsequent chemical and physical soil analysis, another part was kept moist and was frozen for subsequent soil microbial analyses (MBC, MBN or respiration). This is now clarified in the text.

L139: More information of the incubation is appreciated. How long the samples were incubated before sampling? What was the temperature? Did you sample for only once or multiple times?

Samples were preincubated at room temperature for one week (7 days), measurement was conducted for 24 hours at an interval of one hour. The information was added to the text..

L146: Please give more information of linear regressions. For example, indicate that they only have one predictor. Did you check the normality of residues?

We added respective information. Normality of residues was checked.

L147: and after: What are the reasons for performing mixed-effect models? Why parent materials, texture group and land use are selected as random effect variables? In general, random effects are used when samples are only a small subset of the group or when limited groups are included. Does it aim to make predictors on a larger scale using the limited dataset?

West, Welch, & Galecki suggest in their book "*Linear mixed models*" that such models can be applied to clustered data. We decided to use mixed effect models to capture the effect of soil properties applied as fixed effects. Developed on different parent material or under different land use management the soils showed a further source of variability. Furthermore, we selected these variables as random factors. It is aimed to remove their bias from the specific levels of the applied random factors. To this end, sampling sites were specifically selected that covered the factors parent materials, texture group and land use.

L162-163: Why was response variable transformed but not predictors?

Transformation of the response variable is common and was applied to achieve a normal distribution of the residues. The predictors of the mineral phase determine the variability of SOC and its labile fractions. Therefore we tried to keep them as they occur in the environment/our dataset. SOC (or its labile fractions) as variable part of the soil was consequently transformed to achieved normally distributed residues.

Result

Overall: The readability can be improved by dividing this section into a few subsections due to a large content in this section.

Thanks for this valuable advice, we divided the results in subsections.

3.1 Soil properties and cluster identification

3.2 Bivariate relationships of mineral phase and SOC and its labile fractions

3.3 Estimation of SOC and its labile fractions by mixed effect models

3.4 Comparison of total and local explained variability.

L170: What are "soils and topsoil properties"? Consider revising.

Line was revised. Topsoil was separately mentioned due to the fact that our study is focused on agricultural topsoils. To avoid confusion or misunderstanding we decided to use only the term 'Soil properties'.

L177-178: Are they significantly different or different by looking at means/ranges?

Differences were mostly statistically significant differences. Here we solely wanted to mention that a higher proportion of organic substance was found in grassland soils compared to arable soils.

L190: "Somewhat different" is vague.

It was changed accordingly.

L205: and after: This paragraph is comprised of isolated points, which makes it difficult to follow. A suggestion is to describe Table 3 in a well-organized way to shorten this paragraph. For example, you can follow the order of entire dataset --> land use --> parent materials --> texture, or you can introduce them by the types of predictors. Also, focusing on your key findings helps.

We will try to rephrase it paragraph. Anyhow we include subsections.

L207-208: The items "global cluster" and "local cluster" are explained here but they appear in previous parts (e.g. L18 and L193). Please give explanations when they appear for the first time.

Thanks for this hint, we mention the definitions earlier.

L208 and L94: Please be consistent for "parent material" or "parent rock material".

We changed it. Now it is consistent.

L224: What is "a sufficient extent"? Please specify.

Applying "sufficient" is not objective enough, therefore we rephrased this line and specified it by giving the respective level of $R^2$.

L237 -242: Please indicate that they are from Table 3.

The reference to Table 3 was added.

L240: How to know "weight of samples" is equal? Why does it act as global cluster?

We mean the statistical weight of the samples. We changed the sentence as follows: "The clusters of both land use types largely overlapped and contained a similar proportion of samples from each parent material. Therefore they can be regarded as total clusters.".

L250: It is not clear how to compare R^2 between bivariate regression and mixed-linear model. By the means of each cluster?

Next to the comparison of the explained variance we showed the RMSE to give a measure for model performance. This is now better clarified in the revised text. "By the mixed effect models, $R^2_{cond}$ reach higher explained variance for SOC ($R^2_{cond}$ = 0.39-0.89, RMSE = 0.21 – 0.42%) compared to the bivariate regressions ($R^2$ = 0.00-0.73, RMSE = 0.27-1.12%)."  Further we added some information at section 2.3

L257-258: DCS sites look different from LBS and DLS.

We clarified it. "Models using parent material or texture as random effect mostly showed minor differences for predictions of SOC, HWEC or MBC. Anyhow, for some local clusters (e.g. DCS, LBS and DLS) distinct results were found. Models using land use as random effect were partly distinct, though, indicating the different influence of land use on SOC and its labile fractions (Table 4).
"

L279-286: Please indicate related Tables and Figures. It is hard to follow.
We now refer to the considered Tables.

L282 & L287-288: This gives me a feeling that you are estimating whether the models were well-fitted. If this is true, comparing R^2 does not make sense. Large R^2 means more variation is explained by predictors. Instead, you have to look at the distribution of residue using e.g. root mean square error (RMSE).

It is aimed by this study to show how well the specific models with their specific parameter combination explained the variance of SOC, HWEC and MBC. Therefore, we rephrased this sentence. We agree that in order to show the goodness of the model fit RMSE is the correct measure. We added this information.

Discussion

L304-305: "for the in total very sandy soils …of LBS". Try to revise this sentence.
Sentence was adapted.

L309: "…SOC in soil" --> "in soil"
It was changed accordingly.

L314-315: "ECEC, Ca and Mg are suitable predictors for SOC in this study"; L317-318: "The minor ability of ECEC (Ca+Mg) to explain SOC.." They look like contradictory. Also, I missed a point that whether you are talking about entire dataset or specified cluster. Table 3 showed that the predictions using ECEC and (Ca+Mg) are largely dependent on parent materials and texture cluster. A possible explanation is that DCS soils had more sands and lower pH, so that Ca and Mg do not contribute to SOC stabilization, whereas DLS and PSS soils had higher pH, so that Ca and Mg bridging play a role in SOC stabilization (see your cited paper). Please consider re-organizing this part.

We clarified these sentences. (L314-315)"The minor ability of ECEC and $(Ca+Mg)_{ECEC}$ and the higher ability of pedogenic oxides to explain variance of SOC and its labile fractions indicated in this study for several cluster (total and local) by bivariate regressions (Table 3), corresponds to findings of Rasmussen et al. (2018)."……

(L317-318)" Ability of ECEC and $(Ca+Mg)_{ECEC}$ was further strongly dependent on the observed parent material or texture cluster. By the mixed effect models, $(Ca+Mg)_{ECEC}$ were more frequently identified as relevant to explain SOC and its labile fractions. Thereby it is shown that by a collective approach of several soil parameters more driver explain a larger part of the variability than by bivariate approaches. As example ECEC and $(Ca+Mg)_{ECEC}$ was found as relevant for the clusters of DLS and PSS, while for DCS it show a minor importance."

L328-333; Grassland had higher SOC contents than arable land, but the PCA showed that they were largely overlapping. This is a good point for discussion. Some explanations will be appreciated.

We amended the Discussion accordingly. "In comparison, mineral phase soil properties clearly separate the dataset while composition of SOM was less enabled for this purpose. Consequently, a broad scatter of the land use clusters was obtained by PCA, suggesting to treat the land use clusters as total datasets as well."

L334-336: "Several studies with…" has only one citation?
Thanks for this hint, we added further studies.

L351-352: Previous explanations are good reasons for using multiple parameter models. However, the reasons for using mixed-effect linear model are not well mentioned. For example, why not multiple fixed-effect model or partial least square regression? My recommendation is to stay in a safe way.

As mentioned above, West, Welch & Galecki suggest in their book "*Linear mixed models*" this type of model for clustered data. Soil parameters (e.g. pedogenic oxides, texture) have an influence (with differing strength) on SOC, HWEC or MBC. Furthermore, there is an effect by factors such as the parent material or land use. To further capture this effect, we decided to use mixed effect models.

L373-374: To be prudent, I would say models of parent materials explained more variation of SOC because we don't if the model-fitting was better than others (see comments on L282). The same for L374-375.
We changed the sentences to highlight the explained variance.

L379 and after: A major finding of this study is that the overall explained variance decreased in the order SOC>HWEC>MBC. Some explanations for this would be appreciated.
Ok, we added further explanations.

L395: Please be consistent with "mixed effect model" and "mixed parameter model".
Ok, thanks for this hint. The relevant line was changed to mixed effect model

Figures and Tables:
Fig. 1 The clustered locations of the four parent materials are likely to introduce differences in topographical and climate conditions. For example, DCS and LBS sites are mostly located on the top of the mountain/hill, whereas PSS sites are located in a flatter area. The difference may affect soil formation and SOC accumulation. Also, the different altitudes between DCS and PSS sites may cause differences in climate conditions. Therefore, it is possible that the variation caused by climate and topography factors was explained by parent material or land use in this study. I just wonder whether something has been performed in experimental design, statistics or anything else to deal with this problem.

It was aimed by this study to estimate the effect of the mineral phase. Selected sampling region covers soils with identical land use and similar climatic/pedoclimatic condictions but the parent material is substantially different. Consequently soil mineral phase properties differ largely between the local sampling clusters. Our prior aim was to show that SOC of local cluster is better explained by local models, at larger scales we fully agree that differences in pedoclimatic conditions were factors needed to explain SOC (and its labile fractions). For local areas the climatic factors is generally not relevant independent of where they are.

Table 1: What does the unit for respiration mean? As suggested for L146, more information of the incubation is needed.

There was a typo in the unit. We corrected it: [$\mu g$ $CO_2$-C/(g dry matter h)]
We added more information regarding the incubation.

A suggestion for Fig. 2: Why not combining Fig. 2 and Fig. S1 if you want to show the readers that parent material and soil texture make good separations while land use make an insufficient separation?

We tried this option. However, with three plots on one page the readability of the individual plots was poor. So we decided to leave it as is with a focus on the two plots showing differences between clusters

Fig. 2 and 5: The shape of the font might be improved as some of them are narrow but others are wide.

Ok, we adapted the fonts of these figures.

Fig. 3 It looks like that the residue of MBC is less normally distributed compared to SOC and HWEC. Particularly, MBC in grassland soils is underestimated. Also, HWEC has a similar but less obvious trend. My questions are: (1) Is the model prediction of MBC less reliable than others due to the skewed distribution of residue? (2) Are there any reasons for the underestimation of MBC in grassland soils?

For SOC, HWEC and MBC there is a trend of underestimation for grassland sites which increases from SOC to the labile fractions. We assume that additional soil properties (e.g. content of fine root biomass) affect the organic matter here. Since this study is focused on mineral phase parameters and did not consider further biological properties models were less suited to explain SOC and its labile fractions in grassland soils.

Fig. 4 What do "dataset", "DCS", "sand" and "arable" on the left mean?

It shows which cluster is shown there with its models. We added information to the figure in order to clarify it.

Table 5: Does the "model" before "global model to local cluster" mean local model?

Yes, this model means the $R^2$ which is received by the consideration of predicted vs measured data of the models for the specific clusters/datasets. We clarified it

Fig. 6: Is it a part of Table 5? Is there any reason to make it a new Figure? Maybe try to combine Table 5, Table S3 and Fig. 6 into a good shape, or move unnecessary information to supplementary.

Fig. 6 shows the performance of the total model and the respective local model, when both are applied to the same local dataset. This was tested by comparing measured and modelled data based on simple linear regression. This yielded a pseudo $R^2$. This information is contained in Table 5. We added information regarding RMSE at Table 5 and Figure 6.

---

## Author Response (AR2)

**Authors response to review submitted on 06 Dec 2021 for soil-2021-81**

Dear authors,

Thanks a lot for considering the feedback I gave on the initial version of your manuscript. You have incorporated most of my comments adequately, but I would like you to clarify a couple of things in the manuscript:

The authors are thankful for your additional advice to improve the quality of our manuscript. We clarified the suggested points.

- With respect to the title: how about '[...] mineral phase characteristics', instead of 'mineral phase parameters'?
It is a good option, we decided to change it according to your suggestion.

- The Discussion section is still one uninterupted text. The readibility would be increased considerably by splitting this into subsections.
Agreed, subsections were added to the discussion

- With respect to my comment to line 41, about POC: would be good to include this argumentation in the manuscript, so it's clear to the reader why POC was not studied
Thank you for this valuable hint, we added the explanation to these lines.

- With respect to my comment to line 108-109, about the abbreviations: I leave this up to you, but I think the readibility of the text will be increased substantially by using more intuitive names for your study sites
We decided to keep the abbreviations even if they are perhaps not fully intuitive. The full terms would increase the length of sentences too much. Shorter abbreviations are difficult because letters always appear in several terms.

- With respect to my comment to line 134, about CFE: As CFE is generally performed on fresh soil, to make sure the microbial community is as little disturbed as possible at the time of analysis, I would like to ask the authors to justify performing the analysis on frozen soil (either through citing articles showing that this has little effect on tbe measured MBC, or by providing the data that's not shown). In addition, I would like to ask the authors to mention in the manuscript that CFE was performed on samples that were frozen prior to analyses, this is important methodological information that is currently not mentioned.
According to Stenberg et al. (1998) freezing of soil samples at -20°C does not affect the microflora, so it is a widely accepted method for sample preservation in soil microbiology.

Stenberg, B., Johansson, M., Pell, M., Sjödahl-Svensson, K., Stenström, J., Torstensson, L., 1998. Microbial biomass and activities in soil as affected by frozen and cold storage. Soil Biology and Biochemistry 30, 393-402.

- With respect to my comment to line 205-206: Table 1 describes much more than 10 parameters, so this is not clear.

We added the information that only 10 from 23 parameters were selected, further we added information to Table 1 indicating which parameters were chosen.

- With respect to the previous formulation of models of 'sufficient extent': Would be good to clarify in the Material and Methods section what you consider a sufficiently good model

Classification of explained variance regarding their quality is for examples given by Cohen (1988) or Achen (1990). Cohen for example termed explained variance ($R^2$) above 0.26 as 'high'. (For our study, however, an explained variance not much larger than 0.26 is not really high. To stay away from such discussion we focused on a relative assessment of models. If a model had a higher explained variance and a lower RMSE, it was termed as 'superior' to models with lower explained variance. Following the reviewer's advice, we removed insufficient terms of model quality and added additional information to material and methods: 'Both $R^2$ and RMSE were used for a comparative assessment of different models rather than for an absolute valuation.'.

Cohen, J. (1988). *Statistical power analysis for the behavioral sciences (2nd ed.)*. Hillsdale, N.J.: L. Erlbaum Associates.
Achen, C. H. (1990). What Does "Explained Variance" Explain?: Reply. *Political Analysis*, *2*(1), 173–184. doi:10.1093/pan/2.1.173

- With respect to my comment to line 243: This has not been changed at this location in the manuscript, please do so
We rephrased the sentence using a more neutral term.

- With respect to my comment to line 309-310: As you discuss the results of your model in the previous sentences, starting this sentence with 'accordingly' refers to those sentences. Would be good to rephrase this, and make it clear that this statement refers to the article you cite at the end, e.g.: 'For example, Kaiser and Guggenberger showed that ...'
We adapted this sentence to avoid any confusion.

- With respect to my comment to line 342: Please clarify this in the manuscript as well
We added some information to explain what 'multidimensional' means

- With respect to my comment to line 381-382: I would like to ask the authors to change this wording. You cannot assume that a property you didn't investigate contributes to concentration of SOC fraction, and 'explains' the gap in explained variance. However, you can hypothesize this.
Agreed, we changed it accordingly.